# Hyperparameter Transfer with Mixture-of-Experts Layers

**Tianze Jiang** [1]  **Blake Bordelon** [2]  **Cengiz Pehlevan** [3]  **Boris Hanin** [1]

## Abstract

Mixture-of-Experts (MoE) layers have emerged as an important tool in scaling up modern neural networks by decoupling total trainable parameters from activated parameters in the forward pass for each token. However, sparse MoEs add complexity to training due to (i) new trainable parameters (router weights) that, like all other parameter groups, require hyperparameter (HP) tuning; (ii) new architecture scale dimensions (number of and size of experts) that must be chosen and potentially taken large. To make HP selection cheap and reliable, we propose a new parameterization for transformer models with MoE layers when scaling model width, depth, number of experts, *and* expert (hidden) size. Our parameterization is justified by a novel dynamical mean-field theory (DMFT) analysis. When varying different model dimensions trained at a fixed token budget, we find empirically that our parameterization enables reliable HP transfer across models from 51M to 2B total parameters. We further take HPs identified from sweeping small models on a short token horizon to train larger models on longer horizons and report performant model behaviors.

## 1. Introduction

Model and data scaling have led to remarkable and often predictable improvements in performance of pretrained deep learning systems (Hestness et al., 2017; Kaplan et al., 2020; Hoffmann et al., 2022; Bergsma et al., 2025b). Realizing in practice the potential benefits conferred by training large models, however, requires carefully tuning hyperparameters

(HPs) such as learning rate (schedule), batch size, initialization scale, and weight decay (Li et al., 2025; Wen et al., 2025). Directly tuning each HP at large scale is typically impractical. As model and data grow, it is therefore crucial to understand how model architecture, optimizer details, and model scale dimensions (e.g. number of layers and hidden width) *jointly* affect training dynamics.

This motivates the study of HP transfer (Yang & Hu, 2022; Yang et al., 2022; Bordelon & Pehlevan, 2022; Bordelon et al., 2023; Yang et al., 2023b; Bordelon & Pehlevan, 2025; Bordelon et al., 2024; Dey et al., 2025; Bergsma et al., 2025a; Wang & Aitchison, 2025; Wu et al., 2026), a suite of techniques for obtaining performant HPs in large models directly from good HPs found by tuning substantially smaller models. This HP extrapolation, or transfer, is determined through a *parameterization*, a set of theoretically-motivated rules that predict how changing each scaling dimension of interest (e.g. model width, depth) should modify raw HP values.

The foundation for the study of HP transfer when scaling width is the so-called max-update parameterization ($\mu$P), first derived for multi-layer perceptrons (MLPs) and validated empirically for more realistic settings in (Yang & Hu, 2022; Yang et al., 2022). Subsequent works (Bordelon et al., 2024; Dey et al., 2025; Bergsma et al., 2025b) provide parameterizations that yield HP transfer for transformer language model pre-training at scale when model width and depth (jointly) grow. The purpose of the present article is to extend such HP transfer techniques to *sparse* Mixture-of-Experts (MoE) models (Shazeer et al., 2017), which replace the dense feedforward (FFN) modules in standard decoder-only transformers (Radford et al., 2019) with layers where only a *fraction* of weights are activated per token, reducing pre-training and inference FLOPS (see Section 3.1 for our exact setup). Our main contributions are as follows:

**MoE parameterization.** We extend the *Complete*P parameterization (Dey et al., 2025) (developed for dense transformers) to include MoE-specific scaling rules, allowing for scaling up not only depth and width but also the *number* and *size* of experts at fixed active expert sparsity (Section 3.3 and Section 3.2) without re-tuning HPs at each scale.

**Theoretical grounding.** We provide a theoretical justification of our proposed parameterization using dynamical

[1]Operations Research and Financial Engineering, Princeton University, Princeton, NJ, USA [2]Center of Mathematical Sciences and Applications, Harvard University, Cambridge, MA, USA [3]John A. Paulson School of Engineering and Applied Sciences, Center for Brain Science, Kempner Institute for the Study of Natural and Artificial Intelligence, Harvard University, Cambridge, MA, USA. Correspondence to: Boris Hanin <bhanin@princeton.edu>.

*Proceedings of the 43rd International Conference on Machine Learning*, Seoul, South Korea. PMLR 306, 2026. Copyright 2026 by the author(s).

mean-field theory (DMFT) (Bordelon & Pehlevan, 2022). Specifically, we obtain an explicit description of the training dynamics of residual networks with MoE layers in the *simultaneous* limit of infinite width, depth, expert size, and expert count. The evolution of (finite) network summary statistics (e.g. layerwise feature kernels) is therefore automatically consistent across scale. We find a novel three-level mean-field hierarchy: residual stream representations are mean-field over expert outputs, which are themselves mean-field over individual expert neurons. The DMFT analysis justifies several subtle but important properties of our parameterization, such as the transfer of optimal HPs across expert hidden width multiplier (Figure 2, last column).

**Empirical validation of HP transfer.** We systematically verify (over different datasets and MoE sparsity levels) that our parameterization enables reliable transfer on both the initialization scale (standard deviation $\sigma$) and learning rate (LR $\eta$) when varying width, depth, expert count, and expert size (Figure 2 and Figure 4) on a fixed token budget (1B). We verify HP transfer by tuning base models with 38M activated parameters and scale up to (around) 2B total parameters (Figure 2 and Figure 13). During training, we report that stability in MoE pre-training is more sensitive to constant-scale HPs, multipliers that are treated as $\Theta(1)$ in our parameterization (Apdx. D.1). In contrast, balancing constant-scale HPs in dense models can improve performance but does not seem necessary to ensure stability (Bordelon et al., 2023).

**Performant models.** We find empirically that using our parameterization to select HPs leads to strong pre-training performance, even when compared to competitive dense baselines (Figure 1 and Figure 15) when scaling up both model dimensions and increasing total training steps (and thus total number of tokens). Furthermore, our parameterization consistently exhibits uniform expert load balancing even when expert count is scaled (Figure 17).

**Insights on architecture shapes.** We empirically verify, via our zero-shot optimal HP, existing findings (Krajewski et al., 2024; Boix-Adsera & Rigollet, 2025) on scaling expert size versus expert count in MoEs. Without the need to retune HPs at each scale, we consistently recover the benefit of increasing number of experts over expert sizes at fixed parameter count (Figure 5 and Figure 16).

## 2. Related backgrounds

**Mixture of Experts.** Mixture of Experts (MoE) layers represent a paradigm shift in neural architecture design, decoupling parameter count from computational cost via conditional routing, and significantly reducing FLOP at both training and inference. Modern implementations of these architectures, pioneered by (Shazeer et al., 2017) and

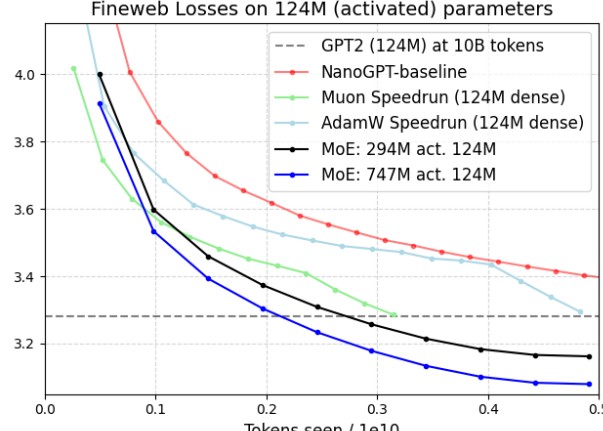

*Figure 1.* Matching active architecture to be GPT2-small (124M) and 500K batch size, comparison of MoE training loss on our zero-shot Adam HPs (found from tuning 38M activated base models) on FineWeb versus (dense GPT) baseline (Karpathy, 2023), speedrun AdamW, and speedrun Muon loss curves (Jordan & contributors, 2025) towards the 3.28 val. loss for baseline at 10B tokens. See Section 5 for details. Dense benchmarks are taken before Oct.14, 24, where more advanced architecture modifications such as zero down-projection init. and QK-norms are applied.

scaled through GShard (Lepikhin et al., 2020) and Switch Transformer (Fedus et al., 2022), utilize a learned gating mechanism to route tokens to a sparse subset of top-k experts as a way to deploy models with trillions of parameters while maintaining per-token computation of much smaller dense models. Recent progress such as Mixtral 8x7B (Jiang et al., 2024), Expert-Choice (Zhou et al., 2022), the DeepSeek-MoEs (Liu et al., 2024; Dai et al., 2024) have further refined architectural details such as routing strategies (e.g., shared experts, loss-free load balancing) to maximize knowledge specialization and training efficiency. However, despite these advances, language model pretraining at scale remains difficult due to challenges such as expert collapse (where the router favors only a few experts), dead or super experts (Su et al., 2025), and training instability (Zoph et al., 2022), often requiring complex ad-hoc strategies to overcome.

**HP transfer.** Traditional approaches to HP tuning at scale require either grid-searching in a large model or extrapolating from power laws fit to a family of smaller models (Li et al., 2025; Zhou et al., 2026). Instead, our approach seeks the direct (rule-based) transfer of optimal HP from small models to larger ones. The core behind transfer is parameterizations that stabilize the forward and backward passes across scales. First studied for scaling the width of MLPs, this stability condition resulted in two types of parameterizations: the *Neural Tangent* parameterization (NTP) (Hayou et al., 2022; Jacot et al., 2020; Roberts et al., 2022) and the *Mean-Field* parameterization (MFP) (Mei et al., 2018; Chizat et al., 2019; Rotskoff & Vanden-Eijnden,

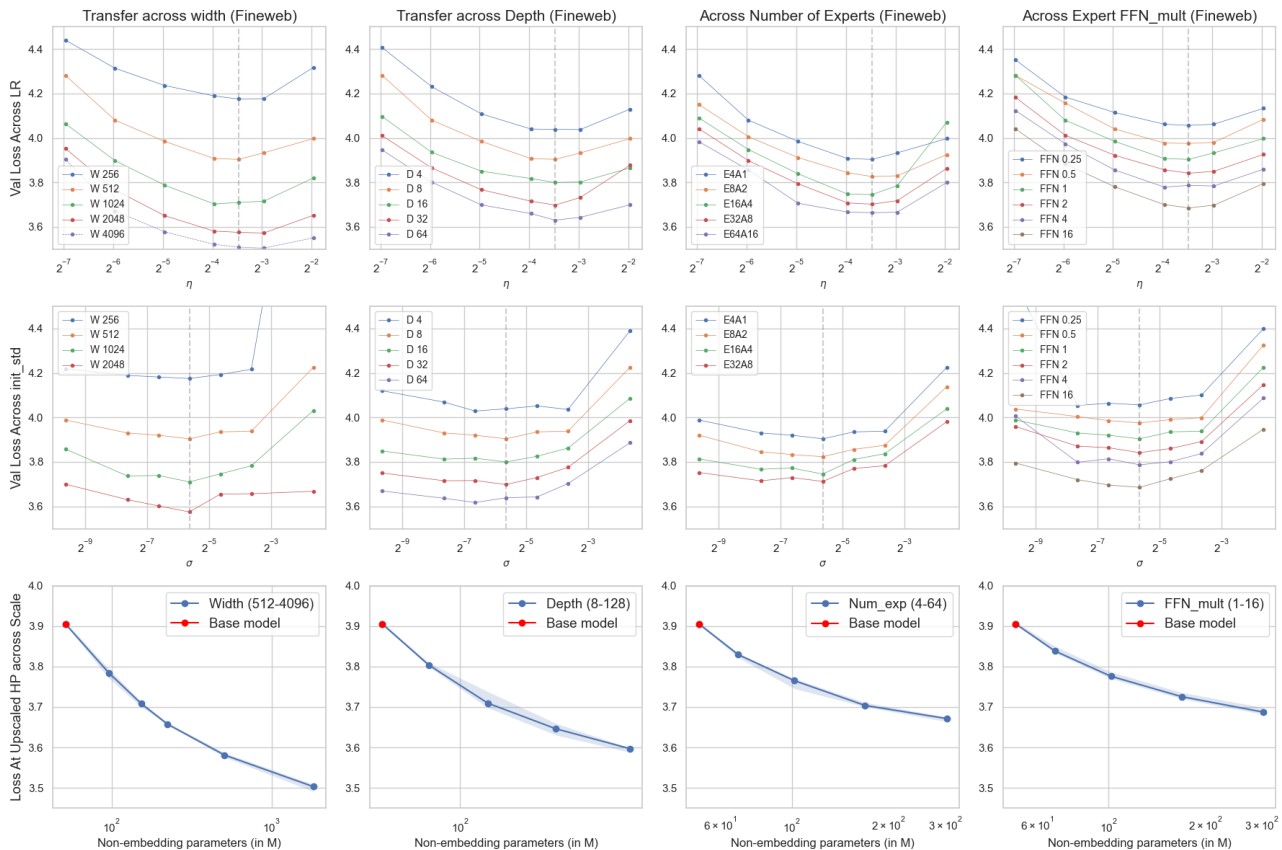

*Figure 2.* Global base learning rate (first row) and global base init (second row) transfer trained on 1B tokens (2000 steps) on the Fineweb dataset, with different model sizes from 20M to 1.8B scaling across width, depth, number of experts (fixing sparsity), and expert MLP hidden multiplier. We fix the base config ($n_{\mathrm{embd}}$ (W) $= 512$, $L = 8$, $(n_{\mathrm{exp}}, n_{\mathrm{act}}) = (4, 1)$ ('E4A1' in the figure), $\alpha_{\mathrm{ffn}} = 1$) and vary one dimension at a time. Error bars in the last row are (max, min, median) over four independent seeds. See Sec. 5 for details.

2022). Such stability conditions have also been studied via the modular norm (Large et al., 2024) and effective field theory (Dinan et al., 2023; Yaida, 2022; Roberts et al., 2022).

The seminal works (Yang & Hu, 2022; Yang et al., 2023a) introduce the concept of a max-update parameterization ($\mu$P), defined by a set of explicit feature learning desiderata satisfied by MFPs and empirically enabling direct *transfer* of good hyperparameters. More recent works (Dey et al., 2025; Mlodozeniec et al., 2025; Wu et al., 2026) extended $\mu$P to allow for depth, width, and other scale parameters in practical-sized (dense) model pretraining with various architectures on practical horizons.

**DMFT** Dynamical mean-field theory (DMFT) provides a way to theoretically analyze HP transfer by showing training dynamics of finite networks to converge to a well-defined limit. This framework traces the evolution (in infinite-sized networks) of mean-field model statistics, such as inner-product kernels for hidden activations and gradients. Earlier DMFT analysis were derived for MLPs (Bordelon & Pehlevan, 2022), Deep ResNets (Bordelon et al., 2023), multi-

head self-attention (Bordelon et al., 2024) and transformers in the infinite-depth limit (Bordelon et al., 2023).

In our present work, while rules for how to set scaling exponents in learning rates and initializations can be identified heuristically by verifying that the entry-wise updates in hidden states of the networks are $\Theta(1)$ (the $\mu$P desiderata) with respect to scaling variables, to prove that these rules actually lead to models converging to a stable infinite limit, we need to verify that there is a well defined dynamical system that no longer depends on exact scaling variables, which leads the necessity of DMFT analysis (Bordelon et al., 2023). In this work, for instance, we use the DMFT to show that there is indeed a limit that does not depend on number of experts but only on activation sparsity. This implies that training dynamics should be consistent across all scaling parameters for a fixed sparsity. Our DMFT analysis also indicates that the limiting dynamics do not depend on the FFN ratio (expert size), which provided theoretical support for HP transfer across FFN ratio. These conclusions *cannot* be justified through heuristic $\mu$P dimensional analysis alone.

**Concurrent work.** Independent and concurrent work (Malasnicki et al., 2025) also studies LR transfer in MoEs as the embedding width scales. However, our contributions go beyond in several ways: **(1)** We study transfer of not only base LR but also init. scale, on not only width but also on depth, number of experts, *and* expert sizes, while reporting a full suite of auxiliary empirical details, and **(2)** we rigorously back our parameterization with a novel DMFT analysis (Section 4) that goes beyond calculating $\mu$P gradient heuristics, demonstrating novel theoretical insights.

After the first version of this work, we noticed independent and concurrent work (Vankadara et al., 2026), which (also) studies the DMFT limit of MoE models under a fixed sparsity. The main difference between our studies is that the notion of a scale-invariant limit in (Vankadara et al., 2026) is somewhat different from ours, and hence the necessity of a more sophisticated analysis in their regimes.

## 3. Scaling recipe for MoEs models

We begin by detailing in Section 3.1 the MoE transformer architecture used in our experiments. We then briefly justify in Section 3.2 why we fix expert sparsity across increasing model size. Finally, in Section 3.3 we adapt ideas from $\mu$P (Yang & Hu, 2022) and *Complete*P (Dey et al., 2025) to provide a heuristic derivation of our proposed parameterization, the set of scaling rules for adjusting HPs as function of (growing) model dimensions.

### 3.1. Model setup

Our theory and experiments concern decoder-only Transformer language models (Radford et al., 2019) with MoE modules in the feed-forward (FFN) layers. In this architecture, a sequence of $T$ input tokens is mapped to an output by first up-projecting (along with positional embedding) each token to a sequence $h^{(0)}$ in $\mathbb{R}^{n_{\text{embd}} \times T}$. This initializes the residual stream, whose updates are computed through $2L$ residual layers that intersperse $L$ pre-LayerNorm multi-head self-attention (MHSA) blocks (for $\ell = 0, \ldots, L-1$):

$$h^{(2\ell+1)}[0{:}T] = h^{(2\ell)}[0{:}T] + \frac{1}{L}f_{\text{MHSA}}(\text{LN}(h^{(2\ell)}[0{:}T]))$$

with $L$ pre-LayerNorm MoE layers (applied position-wise)

$$h^{(2\ell+2)} = h^{(2\ell+1)} + \frac{1}{L}f_{\text{MoE}}(\text{LN}(h^{(2\ell+1)})).$$

An un-embedding layer down-projects the final residual representation $h^{(2L)}$ to produce an output. The $1/L$ multipliers on residual block outputs are chosen following results in (Dey et al., 2025), which give a parameterization called *Complete*P that ensures HP transfer when scaling depth and embedding dimension in dense models. Our parameterization keeps *Complete*P unchanged when scaling depth and

width but allows for separately scaling both expert width and number of experts in the MoE module

$$f_{\text{MoE}}(h) \triangleq \frac{1}{n_{\text{act}}} \sum_{i \in A(h)} g_i(h) \cdot E_i(h) \in \mathbb{R}^{n_{\text{embd}}}, \quad (1)$$

where $g_i(h) \triangleq \sigma(r_i) \in \mathbb{R}$ are (un-normalized) mixing coefficients, where $\sigma$ is a non-linear (sigmoid) function, and

$$r_i \triangleq \left(W_{\text{router}}^{(i)}\right)^T h \in \mathbb{R}, \quad W_{\text{router}}^{(i)} \in \mathbb{R}^{n_{\text{embd}}}.$$

The activated set $A(h) \subseteq \{1, \ldots, n_{\text{exp}}\}$ is a token-dependent subset with $n_{\text{act}}$ indices of *activated experts* determined by

$$A(h) = \text{top}_{n_{\text{act}}}(\{g_i(h) + b_i; \; i = 1, 2, \ldots, n_{\text{exp}}\})$$

in which expert biases $b_i \in \mathbb{R}, i = 1, \ldots, n_{\text{exp}}$ are trainable and *only* participate in the hard-routing. We take each expert $E_i$ to be a single hidden-layer MLP

$$E_i(h) = W_{\text{down}}^{(i)} \phi\left((W_{\text{up}}^{(i)})^T h\right) \in \mathbb{R}^{n_{\text{embd}}}$$

with a hidden layer of size $\alpha_{\text{ffn}} \cdot n_{\text{embd}} \in \Omega(n_{\text{embd}})$, and $W_{\text{up}}^{(i)}, W_{\text{down}}^{(i)} \in \mathbb{R}^{n_{\text{embd}} \times \alpha_{\text{ffn}} n_{\text{embd}}}$. Following (Wang et al., 2024; Dai et al., 2024; Singh et al., 2026) as well as a line of open-source models using similar architectures, our setup *disentangles* weights of expert mixing: for each expert $i$, $g_i(h)$ *only* depends on $W_{\text{router}}^{(i)} \in \mathbb{R}^{n_{\text{embd}}}$. Practically, we did not observe significant differences in model performance across scale between different types of mixing coefficients (e.g. softmax weighting, see Figure 18).

**Training.** We consider the standard Adam optimizer (Kingma & Ba, 2017) (see Apdx. C for discussion on weight decay). During training, we treat the activated experts sets $A(h)$ in each layer as no-grad vectors, and router matrices *only* receive gradients through the expert mixing coefficients $g_i(h)$. Writing $\text{Load}_i \in [0, 1]$ for the (batch) proportion of tokens routed to expert $E_i$, we encourage *expert load-balancing*, i.e. we ask that

$$\text{Load}_i \approx \kappa \triangleq n_{\text{act}}/n_{\text{exp}},$$

ensuring that the fraction of tokens routed to each expert is approximately the same. In contrast to a line of prior works (Shazeer et al., 2017; Fedus et al., 2022) which uses an auxiliary loss as regularization to balance expert load, we adapt the auxiliary-loss-free (Wang et al., 2024; Liu et al., 2024) framework, which encourages expert load balancing by (only) directly updating expert biases

$$b_i \leftarrow b_i - \eta_{\text{bias}} \cdot (\text{Load}_i - \kappa) \quad (2)$$

without affecting updates to other weights. See Apdx. D.2 for further discussion around this choice.

*Table 1.* Parameter groups and their scaling rules for the MoE module at each layer as per derivation in Section 3.3. See also Table 2 in the Appendix for a complete version.

| Parameter | Dimension | Init Std | LR |
|---|---|---|---|
| Router | $\mathbb{R}^{n_{\text{embd}} \times n_{\text{exp}}}$ | $n_{\text{embd}}^{-\gamma}$ | $n_{\text{embd}}^{-1}$ |
| Expert bias | $\mathbb{R}^{n_{\text{exp}}}$ | $0$ | $1$ |
| Expert (up) | $\mathbb{R}^{n_{\text{embd}} \times \alpha_{\text{ffn}} n_{\text{embd}}}$ | $n_{\text{embd}}^{-1/2}$ | $n_{\text{embd}}^{-1}$ |
| Expert (dn) | $\mathbb{R}^{n_{\text{embd}} \times \alpha_{\text{ffn}} n_{\text{embd}}}$ | $n_{\text{embd}}^{-1/2}\alpha_{\text{ffn}}^{-1}$ | $n_{\text{embd}}^{-1}\alpha_{\text{ffn}}^{-1}$ |

### 3.2. Scaling that preserves sparsity versus topK

Let us briefly explain and emphasize an important design choice for scaling: we scale up $n_{\text{exp}}, n_{\text{act}} \to \infty$ while preserving sparsity $\kappa = n_{\text{act}}/n_{\text{exp}}$ (which we think of as a small positive constant). This is instead of scaling up $n_{\text{exp}}$ while fixing $n_{\text{act}}$ (Fedus et al., 2022) and sending $\kappa \to 0$.

We have three practical motivations for this choice. First, (assuming perfect balancing) in a batch of $B$ tokens, each MoE expert sees only $\kappa B$ tokens, whereas self-attention and routing see the full $B$ tokens. This suggest HPs will not transfer across different $\kappa$ (partly confirmed in Figure 10). Second, for large deployments, it has been empirically reported that the practical range of $\kappa$ is often hardware-bottlenecked by communication cost (Liu et al., 2024; Step-Fun et al., 2025) and thus bounded below. Finally, a constant $\kappa$ is natural in the mean-field analysis (Sec. 4) as it represents conditioning on a constant-probability event when integrating over the mean-field measure of experts. Our theory suggests that HPs will transfer at fixed $\kappa$ and our empirical results support this point. For applications with a bottlenecked $n_{\text{act}}$, our recipe is still helpful via (a) start with a fixed target sparsity and scale up granularity; *or* (b) fix an expert configuration and scale up width/depth; *or* (c) apply distillation after training a model with more active parameters.

### 3.3. Proposed MoE Parametrization

For each parameter group (e.g. routing weights, up/down projections in each expert, and expert biases), a practitioner must specify two hyperparameters: initialization std. and learning rate. By a parameterization, we mean a set of rules for how, given a set of HPs for a model at one scale to construct a corresponding set of HPs after up-scaling some of the model dimensions depth $L$, width $n_{\text{embd}}$, expert width (governed by $\alpha_{\text{ffn}} \in \Omega(1)$), and expert count $n_{\text{exp}}$.

Prior work, notably (Yang et al., 2022; Bordelon et al., 2024; Dey et al., 2025), derived parameterizations for dense transformers that exhibit HP transfer across model width and depth. We adopt those prescriptions and focus here on adapting them for sparse MoEs scaling $n_{\text{embd}}$ and $n_{\text{exp}}$ at fixed sparsity $\kappa$. That is, we describe how to set initializa-

tion variances and Adam learning rates for router weights $W_{\text{router}}^{(i)}$, MLP expert down/up projections $W_{\text{down / up}}^{(i)}$ and expert biases $b_i$. Our derivations follow ideas introduced in the max-update ($\mu$P) (Yang & Hu, 2022) and later refined in (Bordelon et al., 2024; Dey et al., 2025). Our results (proposed parameterization) are summarized in Table 1.

**Notation.** For any vector $\theta \in \mathbb{R}^k$ we will shorthand $\theta \in \Theta(C)$ to denote $\|\theta\|_2^2 \in \Theta(k \cdot C)$. To derive scaling rules for Adam, we will also use $\overline{\nabla W}$ to denote the normalized Adam update per step $W_{t+1} \leftarrow W_t - \eta \overline{\nabla W_t}$ and use Adam's approximation by SignGD ($\overline{\nabla W} \in \{-1, 1\}^*$) (Bernstein et al., 2018) so $\Delta w = \eta \overline{\nabla w} \approx \eta \cdot \text{sgn}\left(\frac{\partial \mathcal{L}}{\partial w}\right)$ for any weight group $w$. We also denote by $\cos(v_1, v_2)$ the cosine of the angle between two vectors $v_1, v_2$.

**Operationalizing $\mu$P and *Complete*P.** A core tenet of the $\mu$P (Yang & Hu, 2022) and *Complete*P (Dey et al., 2025) approach to HP transfer is to choose init. and LR so that components of network pre-activations and residual blocks are $O(1)$ at initialization and are updated by $\Theta(1)$ in every training step. To operationalize this for MoEs, we require a stronger condition where max-update conditions have to also hold for *each* individual expert component (mixing co-efficient and expert output). We schematically denote by $z$ the MoE layer output $f_{\text{MoE}}$, individual expert output $E_i(h)$, expert hidden activations $h_{\text{up}} \triangleq (W_{\text{up}})^T h$, or mixing coefficient $g_i(h)$ for $i = 1, 2, \ldots, n_{\text{exp}}$. Viewing these as functions of the MoE parameters $W$, our heuristic requires that the change in $z$ from the corresponding $W$ *separately*:

$$\eta_W \overline{\nabla W} \frac{\partial z}{\partial W} = \Delta W \frac{\partial z}{\partial W} = \Theta(1). \quad (3)$$

To derive from conditions of the form (3) to our parameterization, we will often make so-called "Law of Large Numbers" type alignment assumptions (Yang & Hu, 2022; Yang et al., 2023b; Everett et al., 2024): if $v$ is either a vector of pre-activations or their change after one step of training, and $w$ is either a the vector of model weight updates after one step of training or model weights (*respectively*), then their dot products scales like:

$$v^T w / \|v\| \|w\| = \cos(v, w) \in \Theta(1). \quad (4)$$

whereas two random $d$-dimensional vectors typically give alignments of $\cos(v, w) \in O(1/\sqrt{d})$.

**Router weights.** We start by deriving the parameterization for the MoE router matrix $W_{\text{router}}$. We apply (3) on $W = W_{\text{router}}^{(i)}$ and $z = g_i(h) = \sigma\left(h^T W_{\text{router}}^{(i)}\right)$. Assuming that $\sigma'(\cdot) \in \Theta(1)$, we arrive at our first desiderata:

**Desideratum 1.** *We want for each $i = 1, \ldots, n_{\text{exp}}$ that*

$$h^T \cdot \Delta W_{\text{router}}^{(i)} = \Theta(1).$$

*where $h \in \Theta(1)$ from pre-LayerNorm.*

Recall $\Delta W_{\text{router}}^{(i)} = -\eta_{\text{router}} \overline{\nabla W_{\text{router}}^{(i)}}$. Assuming an LLN-type alignment between $h$ and $\Delta W_{\text{router}}^{(i)}$, and noting that $||\Delta W_{\text{router}}^{(i)}||, ||h|| \in \Theta(\sqrt{n_{\text{embd}}})$ yields

**Scaling Rule 1.** *The router matrix $W_{\text{router}}^{[n_{\text{exp}}]}$ has learning rate $\eta_{W_{\text{router}}^{[n_{\text{exp}}]}} \in \Theta(1/n_{\text{embd}})$.*

At initialization, $z = g_i \in O(1)$ requires $W_{\text{router}}^{(i)} \in \Theta\left(n_{\text{embd}}^{-\gamma}\right)$ for $\gamma \geq 0.5$ (again, when $\Delta z \in \Theta(1)$, it is not necessary to initialize at $\Theta(1)$). Empirically, we observed no practical impact of the router initialization scaling on the loss, so long as it is not numerically too large. In the literature, (Shazeer et al., 2017) used $\gamma = \infty$ (zero router initialization) to ensure load balancing at step 0, (Lepikhin et al., 2020) used $\gamma = 1/2$ such that router logits are $\Theta(1)$ at init, and (Malasnicki et al., 2025) argues for $\gamma = 1$, which mimics the final layer of mean-field MLPs. All experiments are done with $\gamma = 1$ in the main text.

**Expert biases.** For expert biases, we only need to track how much each update shifts the activated experts set. While $b_{[n_{\text{exp}}]}$ only participate in hard-routing, the spirit of (3) can be carried via tracking $f_{\text{MoE}}$ update by updating $b$ alone:

$$\Delta_b f = \frac{1}{n_{\text{act}}} \left[ \sum_{A_{t+1}(h) \setminus A_t(h)} g_i E_i - \sum_{A_t(h) \setminus A_{t+1}(h)} g_i E_i \right]$$

which leads to (assuming $g_i, E_i \in \Theta(1)$):

**Desideratum 2.** *For each step of update on $b_{[n_{\text{exp}}]}$, the activated set of expert shifts by at least $|A_t(h) \setminus A_{t+1}(h)| \in \Theta(n_{\text{act}})$ (for fixed $W_{\text{router}}$).*

We defer the justification of $\eta_{\text{bias}}$ to Appendix B. As before, max-update principles place no requirement on initialization so long as $b \in O(1)$. However, taking into account expert load balancing, we conveniently initialize biases at zero so no one expert disproportionately receives tokens at init.

**Scaling Rule 2.** *Expert biases $b_{[n_{\text{exp}}]}$ are initialized at zero with LR $\eta_{\text{bias}} \in \Theta(1)$.*

**Expert MLP weights.** Each individual expert module admits a separate 1-hidden layer MLP. Dropping expert indices, define the notations in a forward pass as

$$h_{\text{up}} \triangleq (W_{\text{up}})^T h \in \mathbb{R}^{\alpha_{\text{ffn}} n_{\text{embd}}}; E(h) = W_{\text{down}} \phi(h_{\text{up}})$$

where $W_{\text{up}}, W_{\text{down}} \in \mathbb{R}^{n_{\text{embd}} \times \alpha_{\text{ffn}} n_{\text{embd}}}$. Our goal from (3) is to force $\Delta h_{\text{up}}, \Delta E \in \Theta(1)$ from each step of training by updating individual weight components. Note that

$$\Delta h_{\text{up}} = (W_{\text{up}})^T \Delta h + \Delta(W_{\text{up}})^T h$$
$$\Delta E = W_{\text{down}} \Delta \phi(h_{\text{up}}) + \Delta W_{\text{down}} \cdot \phi(h_{\text{up}})$$
$$= W_{\text{down}} \text{diag}(\phi'(h_{\text{up}})) \Delta h_{\text{up}} + \Delta W_{\text{down}} \cdot \phi(h_{\text{up}})$$

Assuming that $\phi'(\cdot) \in \Theta(1)$, our conditions are:

**Desideratum 3.** *During each step of training update,*

$$(W_{\text{up}})^T \Delta h, \ h^T \Delta W_{\text{up}}, \ W_{\text{down}} \Delta h_{\text{up}}, \ \Delta W_{\text{down}} \phi(h_{\text{up}})$$

*are all in $\Theta(1)$.*

For each individual neuron $j \in \{1, 2, \ldots, \alpha_{\text{ffn}} n_{\text{embd}}\}$, assuming LLN alignment (4) for $h$ and the row vector $(\Delta W_{\text{up}})_j$ sets the up-projection LR in that:

$$\eta \| \overline{\nabla (W_{\text{up}})_j} \| = \| \Delta (W_{\text{up}})_j \|_2 \in \Theta(\|h\|_2^{-1}) = \Theta(n_{\text{embd}}^{-1/2}).$$

To set $\sigma_{\text{init}}(W_{\text{up}})$ we use a similar alignment assumption

$$\|(W_{\text{up}})^T \Delta h\|_2 \approx \|\Delta h\|_2 \|W_{\text{up}}\|_{\text{op}} \in \Theta(\sqrt{\alpha_{\text{ffn}} n_{\text{embd}}})$$

and given that $\|\Delta h\|_2 \approx \sqrt{n_{\text{embd}}}$, our condition requires $\|W_{\text{up}}\|_{\text{op}} \in \Theta(\sqrt{\alpha_{\text{ffn}}})$. The usual scaling from random matrix theory gives:

$$\|W_{\text{up}}\|_{\text{op}} \approx \sqrt{\alpha_{\text{ffn}} n_{\text{embd}}} \sigma_{\text{init}}$$

which determines $\sigma_{\text{init}}$. These conditions also guarantees $\Delta h_{\text{up}} \in \Theta(1)$. For $W_{\text{down}}$, the learning rate $\eta(W_{\text{down}})$ is derived similarly via applying (4) to $((\Delta W_{\text{down}})_j, \phi(h_{\text{up}}))$, and the initialization is given by

$$\|W_{\text{down}} \Delta h_{\text{up}}\|_2 \approx \|\Delta h_{\text{up}}\|_2 \|W_{\text{down}}\|_{\text{op}}$$
$$\approx \sqrt{\alpha_{\text{ffn}} n_{\text{embd}}} \sqrt{\alpha_{\text{ffn}} n_{\text{embd}}} \sigma_{\text{init}}(W_{\text{down}}).$$

Asking that to be in $\Theta(n_{\text{embd}}^{1/2})$, we therefore find:

**Scaling Rule 3.** *For the MLP Experts, we have*

$$\sigma_{\text{init}}(W_{\text{up}}^{(i)}) = n_{\text{embd}}^{-1/2}, \ \eta(W_{\text{up}}^{(i)}) = n_{\text{embd}}^{-1}$$

*and*

$$\sigma_{\text{init}}(W_{\text{down}}^{(i)}) = \alpha_{\text{ffn}}^{-1} n_{\text{embd}}^{-1/2}, \ \eta(W_{\text{down}}^{(i)}) = \alpha_{\text{ffn}}^{-1} n_{\text{embd}}^{-1}$$

*for all $i \in \{1, 2, \ldots, n_{\text{exp}}\}$.*

Note that the dependence of $\sigma_{\text{init}}(W_{\text{down}}^{(i)})$ concerning $\alpha_{\text{ffn}}$ is distinct from the standard fan-in initialization (see Figure 11 for a simple ablation). A heuristic justification (see also (Chizat, 2025)) is to treat $n_{\text{embd}} = 1$ and $\alpha_{\text{ffn}}$ as the intermediate "width" in a standard two-layer MLP under the mean-field parametrization (as opposed to NTP, which yields the standard fan-in initialization).

## 4. Dynamical mean field theory

The existence of a well-defined mean-field limit in the training dynamics is a strong justification of HP transfer (Yang & Hu, 2022; Bordelon et al., 2023; 2024). To theoretically establish that our parameterization enables a scale-invariant limit (and consequently admits HP transfer across jointly

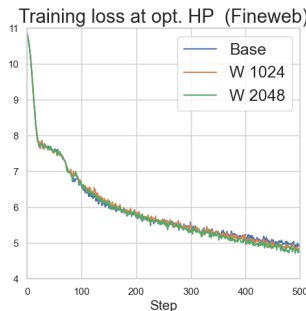 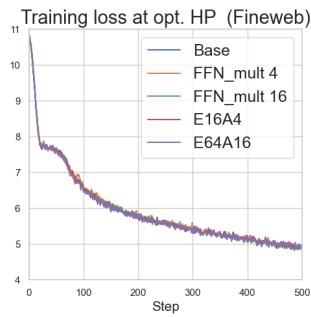 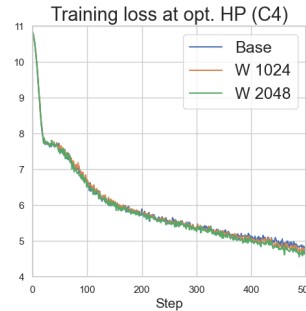 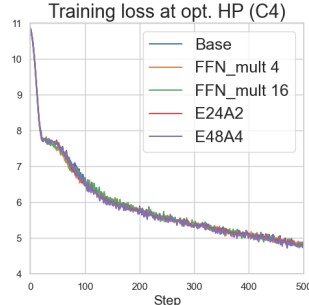

*Figure 3.* (Finding 1.2): Loss curve collapse (scale invariance) of model scaling dimension (in earlier steps) when scaling up the base model in different ways. Parts (1) and (2): Fineweb dataset on sparsity $1/4$. Parts (3) and (4): C4 dataset on sparsity $1/12$.

scaling model dimensions), we outline here a novel approach to studying the training dynamics for a deep residual MoE model[1] in which each input token $x \in \mathbb{R}^D$ is processed by first up-projection $h^{(0)} = W_{\text{embed}} x \in \mathbb{R}^{n_{\text{embd}}}$, then passing through $L$ residual MoE feedforward layers

$$h^{(\ell+1)} = h^{(\ell)} + L^{-1} f^{\ell}_{\text{MoE}}(h^{(\ell)}) \in \mathbb{R}^{n_{\text{embd}}}, \ell = 0, \dots L-1$$

and finally outputting a scalar via the final un-embedding layer $f(x) = W_{\text{unembd}} \phi(h^{(L)}) \in \mathbb{R}$.

Using the analog of our parameterization derived in Section 3.3 for this reduced model, we obtain in Appendix E an explicit mean-field description for the full training dynamics of the model under gradient flow in the limit of diverging residual stream width and expert count $n_{\text{embd}}, n_{\text{exp}} \to \infty$ and constant proportion of activated experts $\kappa = n_{\text{act}}/n_{\text{exp}}$. Our derivation also allows taking the expert size $n_{\text{hid}} \triangleq \alpha_{\text{ffn}} n_{\text{embd}} \to \infty$ and the depth $L \to \infty$ so long as $n_{\text{embd}}/(n_{\text{exp}} n_{\text{hid}} L)$ is bounded in the limit. At a high level, our results reveal an asymptotic three-level mean-field structure made of residual stream neurons, expert outputs, and within-expert neurons:

1. The dynamics of the output $f$ is determined by a finite number of *averages* over residual stream neurons and over the state of experts within each hidden MoE layer.

2. The dynamics of each residual stream neuron depends on the rest of the network only through a finite number of *averages* over all other residual neurons and experts.

3. Sparse expert activation is determined by gating variables $q = \sigma(r) + b$ and set by a quantile threshold $q_\star(\kappa)$ which satisfies $\left[ \mathbf{1}_{q \geq q_\star(\kappa)} \right] = \kappa$. The (hard) gating variables for each expert are thus $\sigma \triangleq \mathbf{1}_{q \geq q_\star(\kappa)} \sigma(r)$.

4. The dynamics of each expert (expert output, routing weights, and bias) depends on the rest of the network

---

[1] While we focus on analyzing a MoE-block only residual stream for the context of this work, we see no technical obstruction to deriving complete DMFT equations for the full sparse transformer architecture by combining with existing analysis of multi-head self-attention layers in (Bordelon et al., 2024).

only through a finite number of *averages* over all other experts, all residual neurons, and all of its own neurons.

5. The dynamics of each neuron within an expert depends on the rest of the network only through a finite set of *averages* over all other neurons in the same expert, all other experts, and all residual neurons.

The averages in the preceding description obey deterministic evolution equations in the large $n_{\text{embd}}, n_{\text{act}}, L$ limit (see Appendix E for a full set of closed evolution equations for macroscopic variables). Several useful observations can be immediately gleaned from the theory:

1. If $n_{\text{embd}}, n_{\text{exp}}, n_{\text{hid}} \to \infty$ with $\alpha_{\text{ffn}} \in \Theta(1)$, the resulting dynamics are independent of the FFN ratio $\alpha_{\text{ffn}}$. This is similar to findings at large depth in dense models (Chizat, 2025) and provides a theoretical basis for transfer across $\alpha_{\text{ffn}}$ as in Figure 2.

2. A large family of joint scalings generates identical dynamics. Let $n_{\text{embd}}(N), n_{\text{hid}}(N), n_{\text{exp}}(N), L(N)$ be diverging functions. The limiting dynamics are universal if $\alpha_\star \triangleq \lim_{N \to \infty} \frac{n_{\text{embd}}(N)}{n_{\text{hid}}(N) n_{\text{exp}}(N) L(N)} = 0$.

3. The depth limit $L \to \infty$ generates a neural ODE for each hidden neuron on the residual stream if $\alpha_\star = 0$ (Bordelon et al., 2024; Dey et al., 2025; Chizat, 2025), but can generate a neural SDE if $\alpha_\star > 0$ (Bordelon et al., 2023; Yang et al., 2023b).

4. $n_{\text{hid}}$ does not need to diverge. If $n_{\text{embd}}, n_{\text{exp}} \to \infty$, the output of the network obeys a deterministic evolution which depends on $n_{\text{hid}}$, but not on $\nu \triangleq \frac{n_{\text{exp}}}{n_{\text{embd}}}$. While this limit is not studied empirically in our work, we point to (He, 2024) for an empirical investigation.

A similar three-level structure can be found for multi-head self-attention (Bordelon et al., 2024), where there exists a measure over attention heads (analogous to our measure over the experts) and an additional measure of key-query neurons within each head (analogous to our per-expert neurons). We also refer to Section 4 in (Bordelon et al., 2023) for a more thorough explanation of how mean-field limit of training dynamics supports reliable HP transfer.

# 5. Empirical results and findings

Our experiments focus on MoE architectures described in Section 3.1, and we defer training details to Appendix A. We consider two different sparsities $\kappa$ on two different natural language datasets: (i) $\kappa = 1/4$ on the FineWeb dataset (Penedo et al., 2024); and (ii) $\kappa = 1/12$ on the C4 (Colossal Clean Crawled Corpus) dataset (Raffel et al., 2020).

In all of our experiments with a fixed token budget, we use a total of (roughly) 1B tokens divided into 2000 batches of 500K tokens and context length 1024. We use an LR scheduler with a linear warmup phase for the first 1000 steps and stable (constant) LR for the latter half. While a typical LR schedule does not warm up for half of the training iterations, our goal with the fixed token budget experiments is to model an early checkpoint or a larger run, which often contains 0.5B (or more) tokens during the warm-up phase.

For each parameter group, we also tune constant-scale multipliers on the learning rate and initialization. Without tuning constant multipliers, we found that (1) nontrivial performance was left on the table and (2) training dynamics (e.g. load balancing loss) can become unstable even at around the optimal HP. See Appendix D.1 for details.

## 5.1. Experiment findings for fixed token budget

**Finding 1.1.** Fixing the sparsity ratio and token budget, optimal base LR and initialization standard deviation transfer across different dimensions of model scale.

Figure 2 and Figure 4 show the main results in this paper: under our scaling rules, relevant hyperparameters are transferred across multiple model dimensions. Furthermore, we also find that the optimal HP identified from small models enables uniform expert load across all experts (Figure 17).

**Finding 1.2.** On the upscaled optimal HPs, loss profiles in early iterations collapse to those of the base model as width, number of experts, and expert sizes increase.

As supporting evidence to Section 4, we demonstrated (Figure 3) that, in our class of scaling models, the training loss profile in early iterations collapses entirely before diverging (where larger models have lower losses). The finding also echoes known results on dense models, albeit for longer training horizons (Bergsma et al., 2025b; Qiu et al., 2025).

## 5.2. Experiments on larger token-horizon

We scale up HPs found from 38M active base model on 1B tokens to longer training horizons under the same batch-size and more steps (while keeping the 0.5B token warmup fixed). Fixing the activated architecture per token (which resembles a standard dense transformer), we can also compare our MoE (with up-scaled HP) training versus known results in dense models (matching active parameter count).

**Finding 2.1.** Optimal HPs found in small models enable stable training on longer token horizons and achieve competitive loss (against active parameter-matched dense models).

Fixing the architecture of the total activated model to match that of the Nano-GPT implementation of GPT2-small (124M) (and having more total parameters), we report a competitive loss curve via zero-shot HPs (Figure 1) compared to checkpoints of the (dense) GPT2-Fineweb speedrun (Jordan & contributors, 2025) under the same batch size. See also Figure 15 for running 7.5B tokens on GPT2-medium (355M activated). For training with a large number of steps, while fixing a stable LR after warmup still yields stable and converging training, including a LR decay (even the simplest cosine decay to zero) empirically improves final validation performance in our experiments.

## 5.3. Tradeoff between expert count versus expert size

We can now study architectural choices such as the tradeoff between expert count and expert size while fixing sparsity and number of (total and active) parameters.

**Finding 3.1.** Increasing number of experts at fixed parameter count improves final model performance.

We justify this via Figure 5 (for short horizon training runs) and further in Apdx D.3 (small models on long horizons). While previous works have reported benefits of more smaller experts (Krajewski et al., 2024; Liu et al., 2024; Boix-Adsera & Rigollet, 2025), our HP transfer results enables such architectural comparison results to be fair (across dimensions) and cheap (without needing to sweep HPs).

# 6. Conclusions and future directions

In this work we derived a novel MoE parameterization (when scaling all of the width, depth, expert count, and expert size) based on the principles of $\mu$P and *Complete*P (Table 1, Table 2, and Section 3.3). All of our parameterization recipe (list of heuristically derived scaling exponents) is then rigorously justified by DMFT analysis (in a slightly simplified setup) by showing that training dynamics are consistent across scale and invariant of the scaling dimensions under a constant sparsity. Finally, we report a full suite of empirical results demonstrating reliable hyperparameter transfer and performant scaling model behaviors, as well as empirical takeaways for training a scaling class of models. Here we will also point out limitations and open directions.

1. As training dynamics in early stages (such as our experiments in 1B token scale) versus later stages (e.g., the "compute-optimal" horizon) can be very different, it would be interesting to expand the HP transfer to

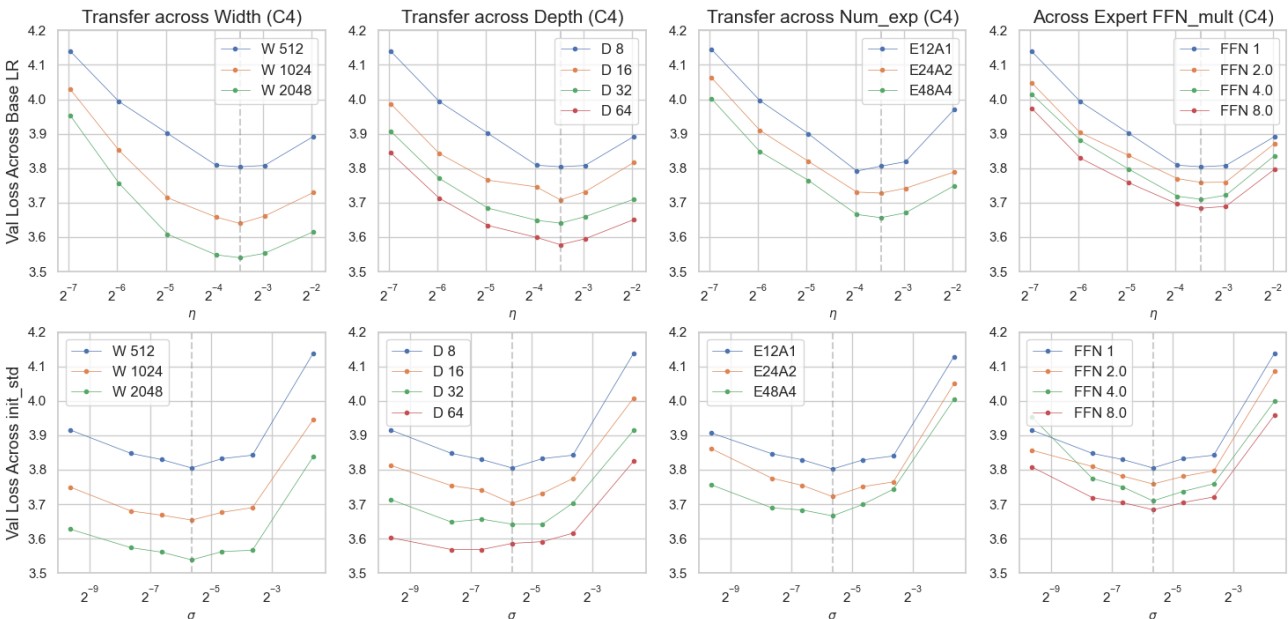

*Figure 4.* Fixed token transfer of global base LR (row 1) and global base init (row 2) on $\kappa = 1/12$ and the C4 dataset

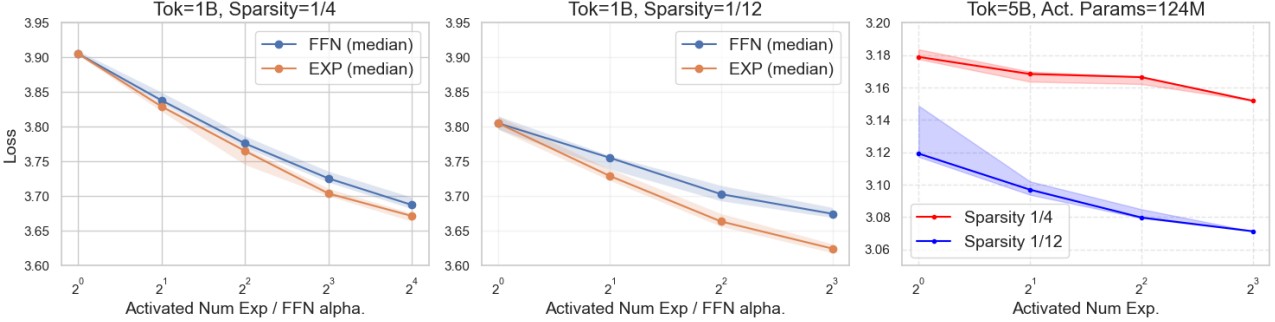

*Figure 5.* Parts (1) and (2): At a fixed token budget of 1B, when scaling up from the base model ($n_{\text{act}} = \alpha_{\text{ffn}} = 1$), increasing the number of experts is more parameter-efficient than expert size. Error bars are (min, median, max) out of 4 seeds. Part (3): At a 5B token horizon with GPT2-small activated and cosine LR cool-down to zero, having more activated experts (and thus inverse proportionally smaller experts) monotonically improves performance. Error bars for $n_{\text{act}} \le 4$ are (min, median, max) out of 3 seeds.

longer training horizons. We also left out discussions of hyperparameters beyond learning rate and init standard deviation, such as batch size, Adam betas, and LR schedules, all of which are interconnected and can have a stronger impact as training horizon expands.

2. While we include finite-dimensional analysis for our DMFT results in Section E, full non-asymptotic DMFT analysis (or HP transfer) is still largely open (see also (Bordelon & Pehlevan, 2023; 2025)). We verify that our theory works practically (e.g. that base $n_{\text{embd}} = 512$ suffices for HP transfer towards the infinite width limit) through experiments and leave the theoretical study of small-width transfer to future work (see also (Ghosh et al., 2025)).

Furthermore, while our techniques in the DMFT analysis extend to SGD or SignGD naturally, a rigor-

ous DMFT treatment on advanced optimizers such as Adam or Muon remain largely open.

3. Finally, we remark that it is not known what "compute-optimal" rules can be practically applied for MoEs (Clark et al., 2022), as (a) Mixture-of-Expert layers achieve significantly better performance compared to dense models under the same FLOP budget, so Chinchilla exponents may not apply, and (b) even when FLOP-matched, MoEs take up significantly more demanding hardware resources due to sparsity, suggesting the necessity of a better way of evaluating compute.

We take our present work on HP transfer as a first step to derive practical yet rigorous scaling laws for MoE models (see (Clark et al., 2022; Krajewski et al., 2024; Li et al., 2025; Zhao et al., 2025) for prior MoE laws in different settings).

## Impact Statement

This paper presents work whose goal is to advance the field of Machine Learning. There are many potential societal consequences of our work, none which we feel must be specifically highlighted here.

## Acknowledgements

T.J. is supported by DARPA AIQ-HR001124S0029. B.B. thanks the the Center of Mathematical Sciences and Applications (CMSA) of Harvard University for support. C.P. is supported by an NSF CAREER Award (IIS-2239780), DARPA grants DIAL-FP-038 and AIQ-HR00112520041, the Simons Collaboration on the Physics of Learning and Neural Computation, and the William F. Milton Fund from Harvard University. This work has been made possible in part by a gift from the Chan Zuckerberg Initiative Foundation to establish the Kempner Institute for the Study of Natural and Artificial Intelligence. B.H. is grateful for support from a 2024 Sloan Fellowship in Mathematics, NSF CAREER grant DMS-2143754, and NSF grant DMS-2133806, and DARPA AIQ-HR001124S0029. We thank Mithril for providing compute resources.

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

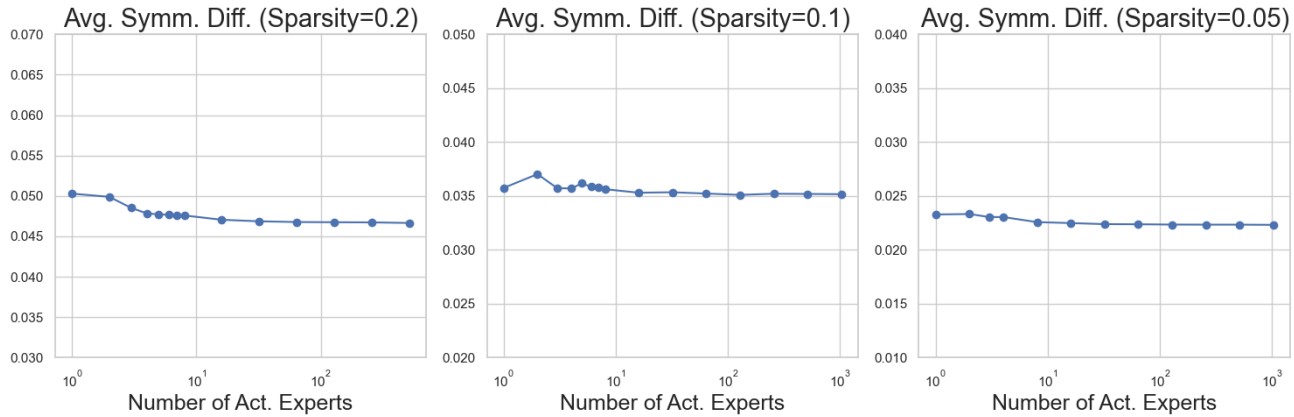

*Figure 6.* (Appendix B) Under random activations, keeping a constant expert bias LR $\Theta(1)$ preserves (across scaling total expert count while fixing sparsity) the average symmetric difference of activated expert proportion per step.

*Table 2.* Completed table of Table 1 for parameter groups and their scaling rules for the MoE module at each layer as per derivation.

| Parameter group (per layer) | Dimension | Init Std $\sigma$ | LR $\eta$ | Adam $\varepsilon$ |
|---|---|---|---|---|
| Embedding and un-embedding | $\mathbb{R}^{V_{\text{vocab}} \times n_{\text{embd}}}$ | 1 | 1 | $n_{\text{embd}}^{-1}$ |
| LayerNorm (scale and bias) | $\mathbb{R}^{2n_{\text{embd}}}$ | 1 | 1 | 1 |
| Attention $W_Q, W_K, W_V, W_O$ | $\mathbb{R}^{n_{\text{embd}} \times 4n_{\text{embd}}}$ | $n_{\text{embd}}^{-1/2}$ | $n_{\text{embd}}^{-1}$ | $n_{\text{embd}}^{-1} L^{-1}$ |
| Router weights $W^{(i)}, i \in [n_{\text{exp}}]$ | $\mathbb{R}^{n_{\text{embd}} \times n_{\text{exp}}}$ | $n_{\text{embd}}^{-\gamma}, \gamma \geq 1/2$ | $n_{\text{embd}}^{-1}$ | $n_{\text{embd}}^{-1} L^{-1}$ |
| Expert bias $b_i, i \in [n_{\text{exp}}]$ | $\mathbb{R}^{n_{\text{exp}}}$ | 0 | 1 | N/A |
| Expert (up) $W_{\text{up}}^{(i)}, i \in [n_{\text{exp}}]$ | $\mathbb{R}^{n_{\text{embd}} \times \alpha_{\text{ffn}} n_{\text{embd}}}$ | $n_{\text{embd}}^{-1/2}$ | $n_{\text{embd}}^{-1}$ | $n_{\text{embd}}^{-1} L^{-1} \alpha_{\text{ffn}}^{-1}$ |
| Expert (dn) $W_{\text{down}}^{(i)}, i \in [n_{\text{exp}}]$ | $\mathbb{R}^{n_{\text{embd}} \times \alpha_{\text{ffn}} n_{\text{embd}}}$ | $n_{\text{embd}}^{-1/2} \alpha_{\text{ffn}}^{-1}$ | $n_{\text{embd}}^{-1} \alpha_{\text{ffn}}^{-1}$ | $n_{\text{embd}}^{-1} L^{-1} \alpha_{\text{ffn}}^{-2}$ |
| All other biases | | 1 | 1 | $L^{-1}$ |

# A. More experiment details

In this section we lay-out some further experiment details from Section 5. We train decoder-only MoE models described in Section 3.1. Our attention, embedding, and normalization setups are based on the public Nano-GPT repo (Karpathy, 2023) and standard GPT-2 style tokenizer (Radford et al., 2019) with a vocabulary size of 50304. For all optimization, we use standard Adam betas $\beta_1, \beta_2 = 0.9, 0.95$ and a negligible Adam $\varepsilon = 10^{-12}$. We use a fixed $d_{\text{head}} = 64$ (while scaling num_head $\propto n_{\text{embd}}$ for large embedding width) for multi-head self-attention following (Dey et al., 2025). We use $QK^T/d_{\text{head}}$ (as opposed to standard $\sqrt{d_{\text{head}}}$) for the normalization of self-attention following (Yang et al., 2022). For experiments with scaling models, our base model has dimensions of base $n_{\text{embd}} = 512$, base $\alpha_{\text{ffn}} = 1$, base $L = 8$, and we up-scale relevant multipliers via the prescribed parametrization in Table 1 (we point our parameterizations for all non-FFN HPs and LN HPs to Table 1 in (Dey et al., 2025)). We train our models using the standard autoregressive cross-entropy loss (i.e. the next token prediction objective) and always report the log perplexity score. As with the standard Nano-GPT (Karpathy, 2023) setup, we use pre-LayerNorm, tied embeddings, absolute learned position embeddings, and GELU nonlinearity. Because router matrices takes up $O(n_{\text{embd}}^{-1})$ fraction of total FFN parameter counts, we may ignore them and count the total parameters as

$$P_{\text{total}} \approx n_{\text{embd}}^2 \cdot L \cdot (4 + 2 \cdot n_{\text{exp}} \cdot \alpha_{\text{ffn}}) + n_{\text{embd}} \cdot (1024 + 50304)$$

where 1024 is the context length and 50304 is the vocabulary size, and for active parameters

$$P_{\text{active}} \approx n_{\text{embd}}^2 \cdot L \cdot (4 + 2 \cdot n_{\text{act}} \cdot \alpha_{\text{ffn}}) + n_{\text{embd}} \cdot (1024 + 50304).$$

Our experiments are run on H100s and are bit-wise deterministic. The entire set of experiments in our paper took around 5000 GPU-Hrs for H100s (the precise number of GPU-Hrs greatly depends on GPU-communication hardware). See the supplementary materials for empirical implementation details.

## B. Learning rate of expert biases

We derive a simple argument on why bias learning rate should not be adjusted when scaling the total number of experts while fixing sparsity. Consider the following pseudo-code, where we assume that the pre-activated $\sigma^{-1}(g_i)$'s are i.i.d. Gaussian, (existing) biases are uniformly distributed in a fixed range, and updated biases are uniformly distributed in a fixed range. This represents the training phase where load imbalance is $\Theta(1)$ proportion of total tokens. The goal is that, regardless of the total number of experts, for a fixed set of gradient distribution, a constant bias learning rate enables constant expert activation set change. See comments below for how we set-up the heuristics.

```
active = sparsity * num_experts
for _ in range(N_TEST):
    logits = 0.02 * np.random.normal(0, 1, num_experts)
    # Assuming random logits, or $r_i$'s, in the main text
    sig_logits = sigmoid(logits)
    bias = 0.05 * np.random.uniform(0, 1, num_experts)
    # Assuming random current expert biases
    mask_top_a = set(np.argsort(sig_logits+bias)[-active:])
    bias_update = 0.01 * np.random.uniform(0, 1, num_experts)
    # Assuming random bias updates from load imbalance
    mask_top_b = set(np.argsort(sig_logits+bias+bias_update)[-active:])
    print(len(mask_top_a & mask_top_b) / active) #symmetric difference portion
```

Results of the above snippet with different sparsities and different number of experts are plotted in Figure 6.

## C. Weight decay

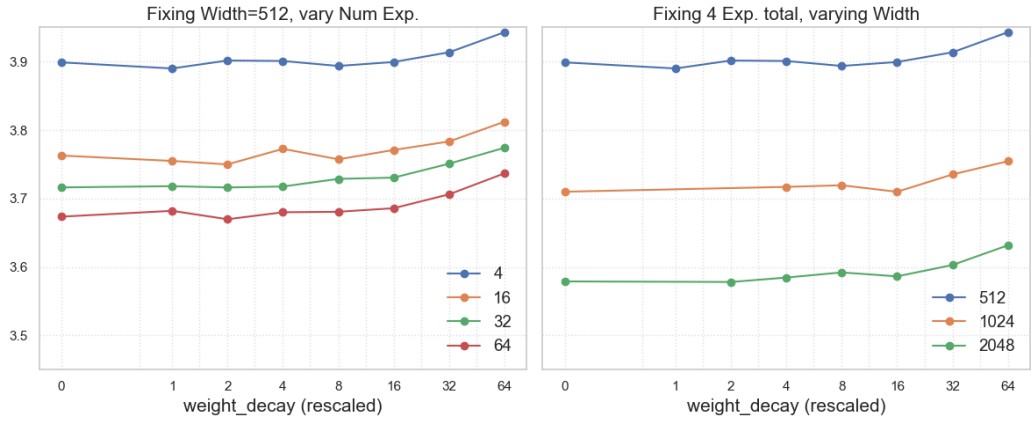

*Figure 7.* Weight decay with independent scaling preserving $\eta\lambda$ (x-axis is the re-scaled $\tau_{\mathrm{EMA}}$). While we see preliminary transfer behaviors across scales when using independent weight decay, we do not observe significant benefit from a nontrivial WD in our scale.

While weight decay (WD) is an important hyperparameter in AdamW (Loshchilov & Hutter, 2019), the exact empirical effect in the context of HP transfer is not well understood. For instance, (Kosson et al., 2025) found that independent weight decay, keeping $\eta \cdot \lambda$ constant (across model scales) such that in every step a constant fraction of weights gets whitened, enables HP transfer. On top of that, (Dey et al., 2025) argued that this independent weight decay $\eta\lambda$ may require depth adjustments according to the depth scaling exponent. However, another line of empirical works (Wang & Aitchison, 2025; Bergsma et al., 2025a) suggests that the scaled invariant should be the effective EMA timescale $\tau = T\eta\lambda$ where $T$ is the number of steps trained. This is contradictory with independent weight decay when $T$ is connected to model parameters, such as width or depth (often the practical case for "compute-optimality" (Kaplan et al., 2020)). Finally, works such as

(Defazio, 2025) suggested WD scheduling, another variable to be considered.

In our experiments with weight decay in a limited scope, we found that at the 1B token horizon with $T = 2000$ steps, a nontrivial weight decay does not significantly outperform having zero weight decay in our horizon, and hence all our experiments in the main text are done without weight decay. See Figure 7.

## D. Other design details and experiment results

### D.1. Constant-scale HP tuning for the base model

Making constant-scale tuning in different types of weight matrices is a standard and studied practice for stability and performance (Everett et al., 2024; Mlodozeniec et al., 2025). Even in dense models, when training is inherently more stable, performance-driven implementations still make efforts to tuning and report benefits. For MoEs, we find that a minimal level of tuning is not only better for performance but also necessary for stability. In terms of HP transfer, it is also more reasonable to scale up from a base model (on which tuning is cheap and efficient) that is both stable and performant at a small scale, as base model misalignments/suboptimality could likely get exaggerated when scaling up (even if the scaling recipe is correct).

In (Yang & Hu, 2022; Yang et al., 2022; Ghosh et al., 2025), the authors pointed out that good HP transfer is crucially based on the "close to optimality" of the scaling. Without specific care, HPs found from tuning small models can be "*contaminated*" by finite-width effects, which fail to yield useful transfer (Figure 17, (Ghosh et al., 2025)). This motivates tuning constant-scaled multipliers on relevant HPs, whose benefits for dense models were remarked in (Everett et al., 2024; Mlodozeniec et al., 2025; Ghosh et al., 2025) that not only promote better loss (in the base model) but likely contribute to more reliable transfer. In our experiments with MoE layers, we found that balancing constant-scaled hyperparameters not only leads to better performance, but it is specifically crucial for stable training dynamics (across different random seeds) on large learning rates. In practice, we enable two constant multipliers for each MoE parameter group for learning rate and initialization. While each group of weights has a three relevant HPs: learning rate, initialization, and multiplier, in the so-called *abc*-parameterization (Yang & Hu, 2022), only *two* degrees of freedom are present. We will take all weight multipliers to be one, so only initializations and learning rates are at play. We report loss results on ablation in Figure 8 and load balancing ablation in Figure 9.

Instead of striving to obtain *the optimal* set of constant-scale HPs across the base model, which is an extremely high dimensional problem and requires exhaustive and expensive tuning even on small models (Mlodozeniec et al., 2025), we only aimed for one *stable* set of constants on the base model that enables consistent model behaviors (for larger learning rates beyond the optimal one) and stable training dynamics (loss across seeds), which we found suffices for HP transfer via our parameterization. The goal is to avoid "cut-off" type behaviors (e.g. Figure 8 without constant tuning, see also Figure 13(b) in (Mlodozeniec et al., 2025)), where loss immediately diverges above optimal learning rate, in which case one is likely transferring model divergence rather than optimal HP, leading to (a) unreliable transfer conclusions because such divergence can happen probabilistically across seed (Figure 8), (b) unsafe HPs to use being closer to divergence limits (albeit optimal on base models), and (c) likely nontrivial performance left on the table. After tuning, we found that our loss dynamics are stable well beyond optimal LRs.

To tune these constants on the base model, we start from the default (all-one) and sweep over each component sequentially on the base model (fixing global learning rate and initialization), similar to coordinate-descent, and only update when significant improvements across seeds are observed. We ended up with `attn_QKV_lr_mult` $= 1/16$, `attn_V_init_mult` $= 1/16$, `router_lr_mult` $= 1/16$, `mlp_down_init_mult` $= 1/4$, `mlp_down_lr_mult` $= 1/16$ (all else being one) in a single pass which gave satisfying base model behaviors already. In our two settings ($\kappa = 1/4$ and $1/12$), we reused the same set of constant-scale HPs, as we found that the constants tuned from $\kappa = 1/4$ worked well to achieve our stability purpose in the $\kappa = 1/12$ base model. In fact, in Figure 10, the degradation of stability happens rather slowly as $\kappa \to 0$ (only at $\kappa = 1/16$ do we see marginally unstable behavior). Finally, while we found success in constant tuning for our purposes, application of normalization layers can also be beneficial (Mlodozeniec et al., 2025). In our experiments, we take the standard normalizations from (Karpathy, 2023) with learning prescriptions from (Dey et al., 2025) and defer the study of advanced normalization techniques into future work.

### D.2. Expert load balancing

We briefly justify our choice of expert load balancing strategy, as opposed to the perhaps more common auxiliary loss type regularization (Shazeer et al., 2017; Lepikhin et al., 2020), where an external penalty $L_{\text{load}}$ is computed (summed over each

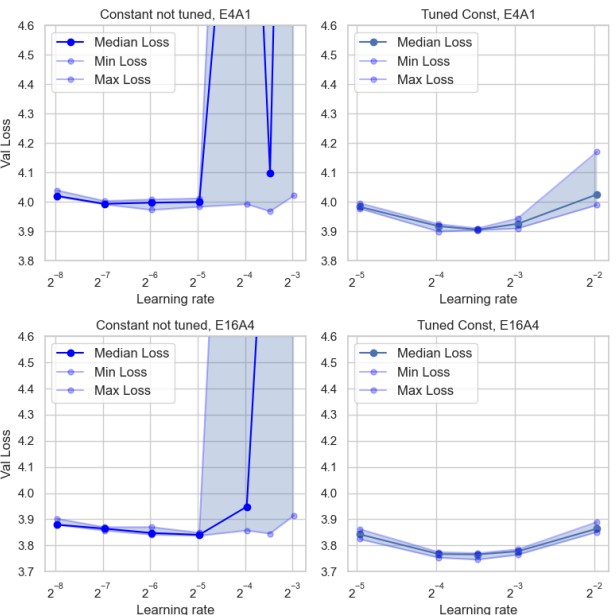

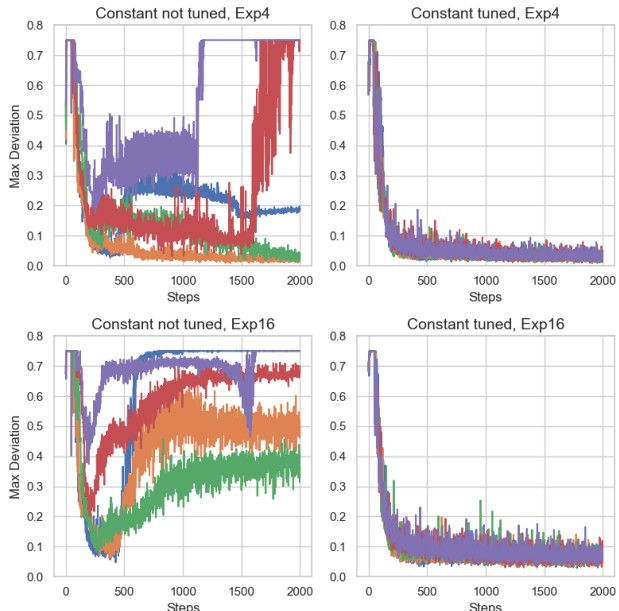

*Figure 8.* Ablation results on constant tuning, run across 4 different seeds with reported (min, median, max) bars. Default constants (1) have higher loss overall, even on the opt LR from sweeping, and (2) on larger LRs, some seeds converge (and even exhibit good performance), whereas others diverge completely, making LR sweep results unreliable. Further issue with untuned constants in terms of load-balancing is in Figure 9.

*Figure 9.* Maximum load deviation in (2) for all experts over all layers (range from 0 to $1 - \kappa = 0.75$) throughout training, where each curve represents a distinct learning rate. We can see that with tuned constants, max load deviation drops very quickly and stays low (across mini-batches), meaning that expert load is balanced effectively. However, for default constants load is not balanced for any learning rate swept. Due to the constant effect of load balancing regularizer in place, a non-balanced expert load can harm model performance and prohibit good transfer.

layer of some load violation function) on top the standard cross-entropy (CE) loss, and the model is trained on

$$L_{\text{total}} = L_{\text{CE}} + \alpha L_{\text{load}}$$

for some $\alpha$. In particular, the object of interest in studying this type of loss is (1) what is a concrete desiderata with respect to load balancing penalty in terms of max-update principles, and (2) whether there exists a suitable parameterization and a fixed scale of $\alpha$ throughout training such that max-update principles will be upheld.

When the backward pass with $L_{\text{load}}$ is involved, all parameters receive a gradient from the regularizer. Therefore, a natural choice could be forcing

$$\nabla_W L_{\text{CE}} = \Theta(\nabla_W L_{\text{Load}})$$

for all parameters groups, so that cross-validation loss and load balancing regularization are on the same scale. However, the practical implications may be more complex as the router matrix at each layer is predominantly responsible for the load balancing at this layer, and it is not clear whether it is desirable or not for FFN or self-attention modules to receive the same gradient norm from load balancing (in other layers) as they would from cross-entropy, or if one router matrix in some layer needs to receive gradients from load-balancing other layers. For instance, should the MLP in Layer 1 be responsible (in terms of gradient scale) for load balancing in Layer 10? Furthermore, when we finally justify HP transfer on training or validation loss, usually the object of study is only concerning cross-entropy and not with the regularized load-balancing loss. So the tradeoff between better validation loss versus strict uniform balance in early training (Dai et al., 2024; Liu et al., 2024) cannot be evaluated fairly without making restricting assumptions.

In conclusion, analyzing a global auxiliary loss requires much more practitioner-specific assumptions (e.g., desiderata in balancing gradients in different layers, desiderata in comparing different models), and that blanket application is not theoretically justified (in fact, aux-loss is not globally minimized with balanced routing). Furthermore, even the empirical application of aux-loss requires scrutiny (e.g. batch-wise vs sequence-wise balancing achieve different objectives (Liu et al., 2024)). While still not a perfect solution, our choice of aux-loss-free balancing is both empirically popular as a replacement

and rids us of having to make much more fine-grained assumptions (e.g. only specific biases receive balancing gradients, and sequence vs batch imbalance losses coincide).

### D.3. More experiments

1. In Figure 11, we run a simple ablation to see that a standard fan-in initialization when scaling $\alpha_{\text{ffn}}$ fails loss-scale invariance, whereas our derived $\alpha^{-1}$ rule on expert down projection layers satisfied the invariance.

2. In Figure 12, we fit runs (with optimal HP) in terms of log parameter count and validation loss at the 1B data scale. We find that a linear fit returns remarkable accuracy. In particular, when fitting against the total number of model parameters, the r-squared coefficient (with validation loss) for total parameters and non-embedding number of parameters are 0.900 and 0.914. When fitting against *activated* number of parameters, total activated r-square is lowered to 0.844 whereas non-embedding fit r-squared slightly increased to 0.933.

3. In Figure 13, we run a few other LR sweeps (with fixed 1B token count) varying different types of hyperparameter configurations scaling expert count and width together. In experiments up to 2.54B models (50x base) and a full 10% loss gap from base model, we see that transfer holds well.

4. In Figure 14, we test transfer on GPT2-small (124M) with 2.5B total number of tokens (20 tokens per activated parameter), with a sparsity $\kappa = 1/4$ similar to the style of Figure 1. Both learning rate and initialization scale transfer holds up at the larger token horizon.

5. In Figure 15, we compare our results ran on GPT2-medium (355M). While recorded (Jordan & contributors, 2025) Speedrun logs for this architecture only exist after the 124M base variant was highly optimized (which vastly improves Figure 1), we still manage to outperform the (tuned) llm.c baseline (Karpathy, 2023) easily.

6. In Figure 16, we see that the tradeoff in Figure 5 (Finding 3.1 in the main text) persists for over-trained models (having 50M and 100M active parameters trained on close to 5B tokens).

7. In Figure 17, we report the maximum expert load imbalance across the entire model. We see that even when we scale up number of experts, the maximum load imbalance across all experts in all layers converges to uniform.

8. In Figure 18, we test and verify that softmax vs sigmoid activation does not significantly (if at all) impact our qualitative transfer observations.

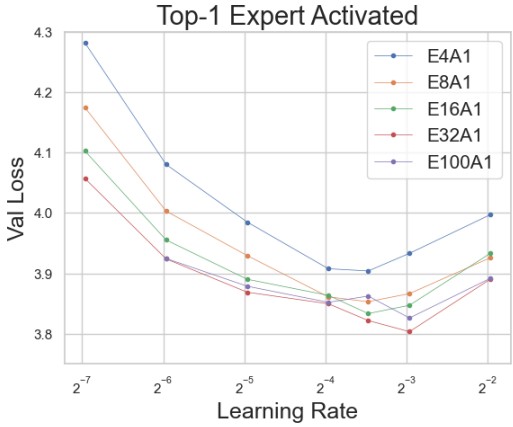

*Figure 10.* Section 3.2: Optimal LR does not transfer in our setting when fixing activating top-1 expert and scaling the number of total experts ($\kappa \to 0$). Constant-scaled hyperparameters (Apdx. D.1) also fail transfer of stability as evidenced by E100A1.

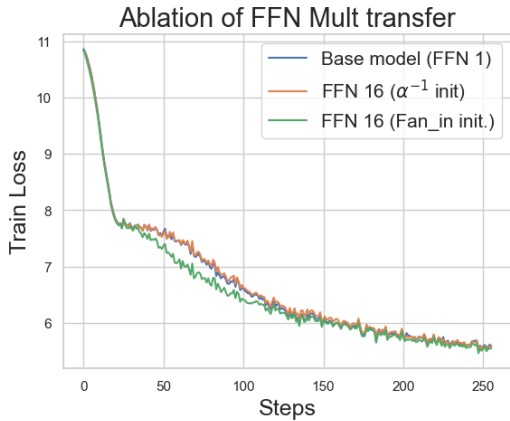

*Figure 11.* The standard fan-in initialization ($\sigma_{\text{dn}} \sim \alpha_{\text{ffn}}^{-1/2}$) for expert down projection fails early-loss scaling invariance when $\alpha_{\text{ffn}}$ scales. See Section 3.3 for our derivation to $\alpha_{\text{ffn}}^{-1}$ and Figure 3.

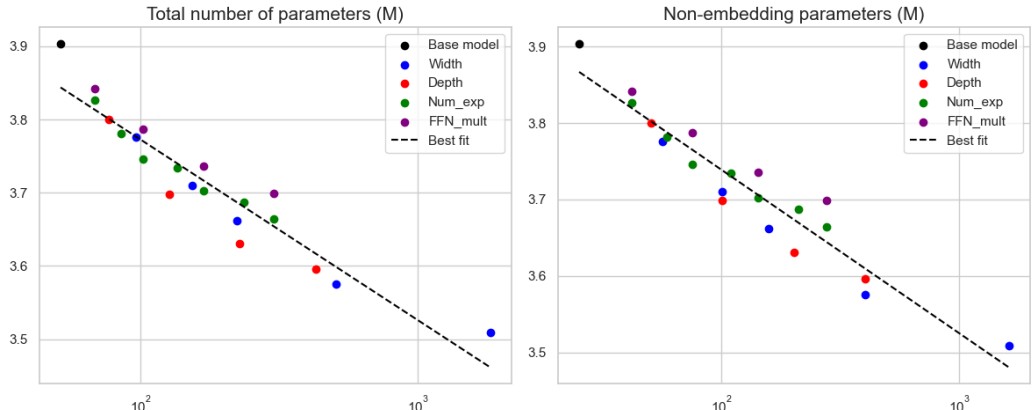

*Figure 12.* We plot model final validation loss versus model size, when scaling different dimensions at the 1B scale (left: total parameters, right: number of non-embedding parameters), we see that the number of non-embedding parameters (in log-scale) is more linearly correlated with final loss.

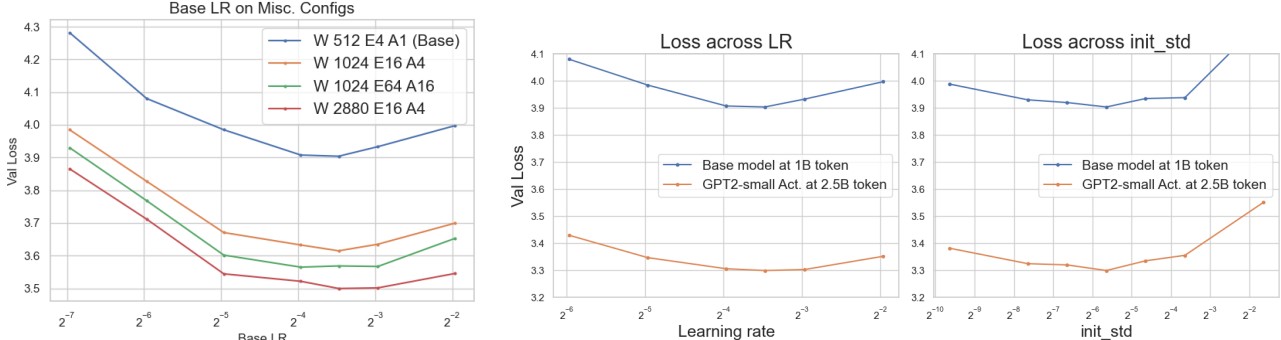

*Figure 13.* Base LR transfer on 1B token when scaling up miscellaneous configurations together (depth 8, $\alpha$1). The largest model taken here has 2.54B total number of parameters and 944M activated.

*Figure 14.* Transfer from base model (1B fixed token) to GPT2-small-124M (with 4 experts, 1 activated, and hidden multiplier 4.0) trained at 20TPP (2.5B token) with no weight decay and cosine LR decay to zero.

# E. Dynamical mean field theory

In this section, we derive the dynamical mean field theory equations for the large width, large expert count, and large depth limit of this model. We assume a deep residual network consisting of $L$ MoE layers without attention. We further focus on gradient flow in the present analysis but this can be easily relaxed to discrete time SGD (Bordelon & Pehlevan, 2022; Bordelon et al., 2023).

**Notation.** For simplicity of notations, we denote in the below:

$$K \triangleq n_{\text{act}}, \quad E \triangleq n_{\text{exp}}, \quad N \triangleq n_{\text{embd}}, \quad N_e \triangleq \alpha_{\text{ffn}} \cdot n_{\text{embd}} = n_{\text{hid}}.$$

We will use $\langle \rangle$ to represent the residual stream neuron average. We will use $[]$ for expert average and $\{\}$ to represent the within-expert neuron average. A covariance kernel will be represented as $C_{ab}^{\ell}(\boldsymbol{x}, \boldsymbol{x}', t, t') = \langle a^{\ell}(\boldsymbol{x}, t) b^{\ell}(\boldsymbol{x}', t') \rangle$ and response functions $R_{ab}^{\ell}(\boldsymbol{x}, \boldsymbol{x}', t, t') = \left\langle \frac{\delta a^{\ell}(\boldsymbol{x}, t)}{\delta b^{\ell}(\boldsymbol{x}', t')} \right\rangle$. Lastly we will use $M_{ab}^{\ell}$ to represent mixture kernels which are expert averages over within-expert variables such as $M_{\dot{\sigma}\dot{\sigma}\mathcal{A}\mathcal{A}}^{\ell}(\boldsymbol{x}, \boldsymbol{x}', t, t') = [\dot{\sigma}^{\ell}(\boldsymbol{x}, t)\dot{\sigma}^{\ell}(\boldsymbol{x}', t')\mathcal{A}^{\ell}(\boldsymbol{x}, t)\mathcal{A}^{\ell}(\boldsymbol{x}', t')]$. Where appropriate, we will often drop indices over $\boldsymbol{x}$ and $t$ and use notation like $\chi^{\ell}\widehat{\chi}^{\ell}$ instead of $\int d\boldsymbol{x}dt\ \chi^{\ell}(\boldsymbol{x}, t)\widehat{\chi}^{\ell}(\boldsymbol{x}, t)$. Our DMFT forward pass notations, as carefully defined below, will be slightly different from Section 3.1 in the main text.

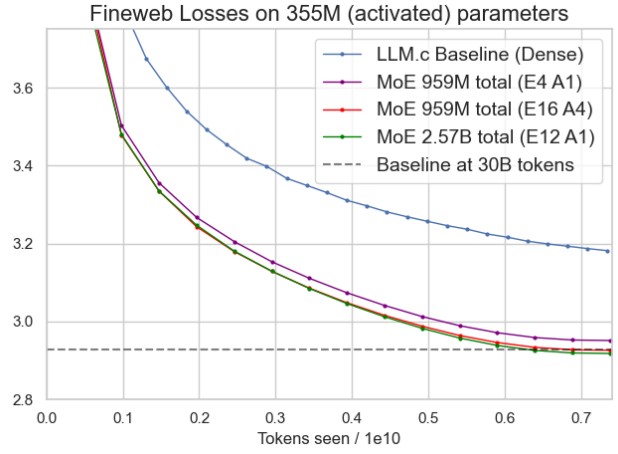

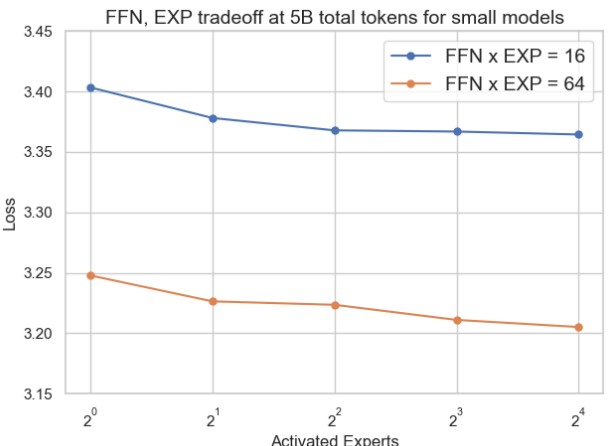

*Figure 15.* Fixing GPT2-medium 355M activated, we compare our scaled MoE models (again zero-shot HP from 38M active base model) under two different sparsity settings (and two different expert size multipliers for $\kappa = 1/4$). Under the same total and activated parameter count, E4A1 significantly underperforms E16A4, which performs similarly to E12A1, despite the latter being larger.

*Figure 16.* Smaller models (51M and 102M activated) trained on a much longer token horizon (4.92B tokens). We see that having a larger number of smaller experts still consistently yield benefit over fewer large experts.

**Universal Dynamics in A Joint Scaling Limit** We will start our analysis by assuming that all variables $N, E, N_e, L$ diverge together with (1) constant sparsity $K = \kappa E$ and (2) the following condition

$$\lim_{N \to \infty} \frac{N}{N_e(N)L(N)E(N)} \equiv \alpha_\star < \infty \tag{5}$$

We will first establish that the DMFT equations in terms of arbitrary finite $\alpha_\star$. However, since most commonly utilized joint scaling protocols result in $\alpha_\star = 0$ we will then specialize to that case. In that case, we show that the limit dynamics follow a universal neural ODE that depends on $\kappa$ but not on the FFN ratio $N_e/N$.

**Defining the Moment Generating Function** Let $f(\boldsymbol{x}, t)$ represent the output of the neural network. We initialize every random initial weight to have unit variance except $\boldsymbol{r}_k$ will be initialized at zero (but the biases are non-zero)[2].

$$Z = \left\langle \exp\left(N \int dt d\boldsymbol{x}' f(\boldsymbol{x}, t) \widehat{f}(\boldsymbol{x}, t)\right) \right\rangle \tag{6}$$

We introduce the definition of the network function $f(\boldsymbol{x}, t)$ and intermediate variables

$$f(\boldsymbol{x}, t) = \frac{1}{\gamma_0 N} \boldsymbol{w}^L(t) \cdot \boldsymbol{h}^L(\boldsymbol{x}, t) \tag{7}$$

---

[2]This will not change the limiting dynamics since the $\frac{1}{N}\boldsymbol{r}_k(0) \cdot \boldsymbol{h}$ will have subleading variance and no response function in the limiting DMFT. Rather, we rely on the random initial biases $b_k(0)$ to generate diversity across experts at initialization.

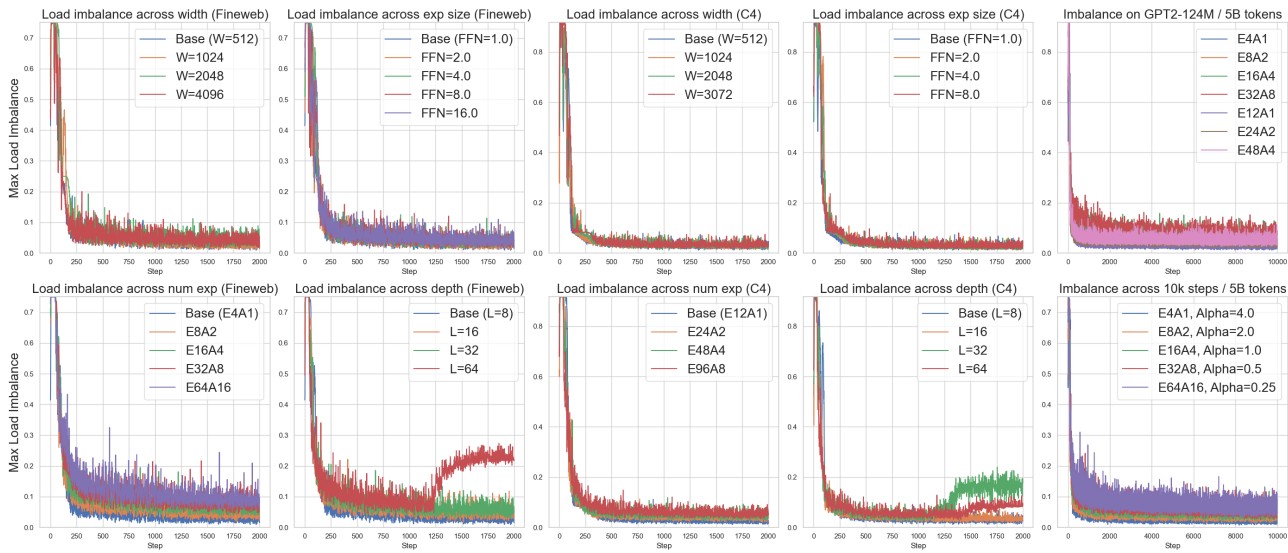

*Figure 17.* Maximum load imbalance in our class of scaling models. Load imbalance (Eqn. 2) is calculated as the max, over $L \cdot n_{\text{embd}}$ total experts in the model, absolute value of (proportion of tokens routed to the expert) subtracts (the target uniform sparsity $\kappa$), similar to Figure 9. Here $y$-axis is from 0 to $1 - \kappa$. The last column shows that load balance convergence persists even when training horizon is extended.

where the hidden features $\boldsymbol{h}^\ell(\boldsymbol{x}, t)$ are determined by the forward pass recursion

$$
\boldsymbol{h}^{\ell+1}(\boldsymbol{x},t) = \boldsymbol{h}^\ell(\boldsymbol{x},t) + \frac{\sqrt{N}}{LN_eE} \sum_{k=1}^E \sigma_k(\boldsymbol{x},t)\boldsymbol{W}_k^{\ell,2}(t)\phi(\boldsymbol{h}_k^\ell(\boldsymbol{x},t))
$$

$$
= \boldsymbol{h}^\ell(\boldsymbol{x},t) + \frac{1}{L}\underbrace{\frac{\sqrt{N}}{N_eE} \sum_{k=1}^E \sigma_k(\boldsymbol{x},t)\boldsymbol{W}_k^{\ell,2}(0)\phi(\boldsymbol{h}_k^\ell(\boldsymbol{x},t))}_{\bar{\chi}^\ell(\boldsymbol{x},t)}
$$

$$
+ \gamma_0 \mathbb{E}_{\boldsymbol{x}'} \int dt' \underbrace{\left(\frac{1}{E}\sum_k \sigma_k(\boldsymbol{x},t)\sigma_k(\boldsymbol{x}',t')C_{\phi,k}^{\ell,1}(\boldsymbol{x},\boldsymbol{x}',t,t')\right)}_{M_{\sigma\sigma C_\phi}^\ell(\boldsymbol{x},\boldsymbol{x}',t,t')} \boldsymbol{g}^{\ell+1}(\boldsymbol{x}',t')
$$

$$
\boldsymbol{h}_k^\ell(\boldsymbol{x},t) = \frac{1}{\sqrt{N}}\boldsymbol{W}_k^{\ell,1}(t)\boldsymbol{h}^\ell(\boldsymbol{x},t)
$$

$$
= \underbrace{\frac{1}{\sqrt{N}}\boldsymbol{W}_k^{\ell,1}(0)\boldsymbol{h}^\ell(\boldsymbol{x},t)}_{\chi_k^{\ell,1}(\boldsymbol{x},t)} + \gamma_0 \mathbb{E}_{\boldsymbol{x}'} \int dt' \underbrace{\left(\frac{1}{N}\boldsymbol{h}^\ell(\boldsymbol{x},t)\cdot\boldsymbol{h}^\ell(\boldsymbol{x}',t')\right)}_{C_h^\ell(\boldsymbol{x},\boldsymbol{x}',t,t')}\sigma_k(\boldsymbol{x}',t')\boldsymbol{g}_k^\ell(\boldsymbol{x}',t') \tag{8}
$$

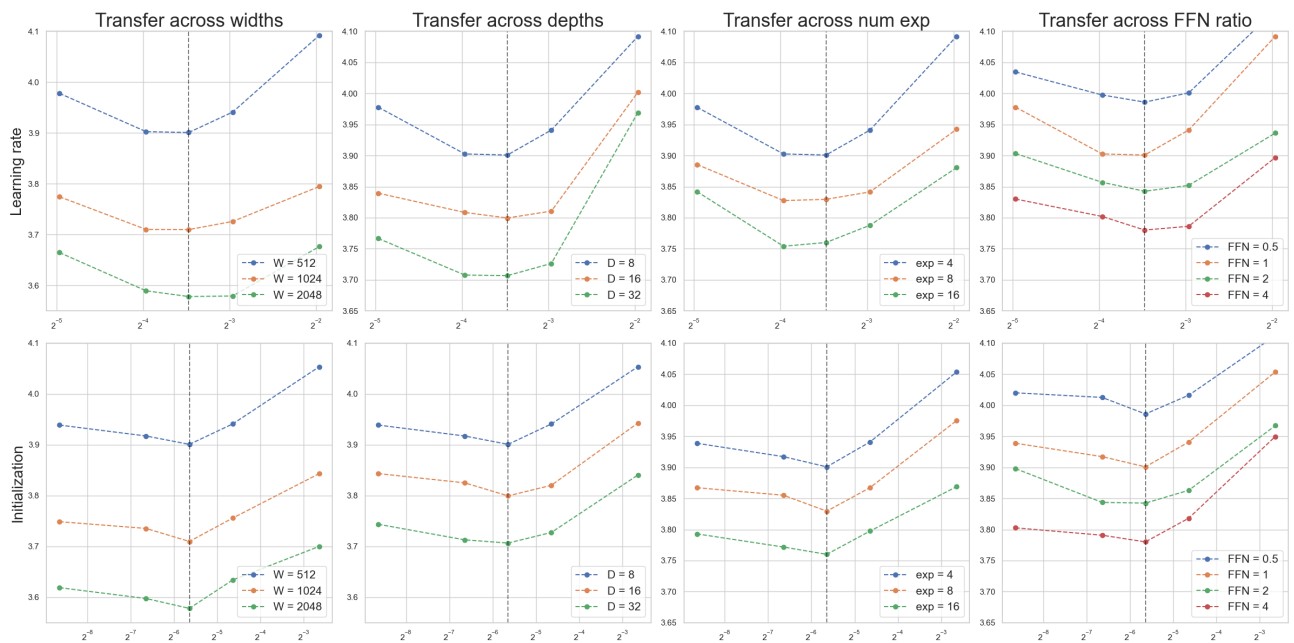

*Figure 18.* Transfer behavior on Fineweb when sigmoid expert mixing is replaced with softmax. Our analysis remains applicable and empirically verified.

Similarly, for the backward pass variables $\boldsymbol{g}^\ell = N\gamma_0 \frac{\partial f(\boldsymbol{x},t)}{\partial \boldsymbol{h}^\ell(\boldsymbol{x},t)}$ we have

$$\boldsymbol{g}^\ell(\boldsymbol{x},t) = \boldsymbol{g}^{\ell+1}(\boldsymbol{x},t) + \frac{\sqrt{N}}{LN_eE} \sum_k \sigma_k(\boldsymbol{x},t) \boldsymbol{W}_k^{\ell,1}(t)^\top \boldsymbol{g}_k^{\ell,1}(\boldsymbol{x},t) \tag{9}$$

$$+ \frac{1}{LE} \sum_k \dot{\sigma}_k(\boldsymbol{x},t) \mathcal{A}_k(\boldsymbol{x},t) \boldsymbol{r}_k(t) \tag{10}$$

$$= \boldsymbol{g}^{\ell+1}(\boldsymbol{x},t) + \underbrace{\frac{\sqrt{N}}{LN_eE} \sum_k \sigma_k(\boldsymbol{x},t) \boldsymbol{W}_k^{\ell,1}(0)^\top \boldsymbol{g}_k^{\ell,1}(\boldsymbol{x},t)}_{\bar{\xi}^\ell(\boldsymbol{x},t)} \tag{11}$$

$$+ \frac{\gamma_0}{L} \mathbb{E}_{\boldsymbol{x}'} \int dt' \Delta(\boldsymbol{x}',t') \underbrace{\left( \frac{1}{E} \sum_k \sigma_k^\ell(\boldsymbol{x},t) \sigma_k^\ell(\boldsymbol{x}',t') C_{g_k}^{\ell,1}(\boldsymbol{x},\boldsymbol{x}',t,t') \right)}_{M_{\sigma\sigma C_g}^\ell} \boldsymbol{h}^\ell(\boldsymbol{x}',t') \tag{12}$$

$$+ \frac{\gamma_0}{L} \mathbb{E}_{\boldsymbol{x}'} \int dt' \Delta(\boldsymbol{x}',t') \underbrace{\left( \frac{1}{E} \sum_k \dot{\sigma}(\boldsymbol{x},t) \dot{\sigma}(\boldsymbol{x}',t') \mathcal{A}_k(\boldsymbol{x},t) \mathcal{A}_k(\boldsymbol{x}',t') \right)}_{M_{\dot{\sigma}\dot{\sigma}\mathcal{A}\mathcal{A}}^\ell} \boldsymbol{h}^\ell(\boldsymbol{x}',t') \tag{13}$$

Further, we have the intermediate gradient fields

$$\boldsymbol{g}_k^{\ell,1}(\boldsymbol{x},t) = \dot{\phi}(\boldsymbol{h}_k^{\ell,1}(\boldsymbol{x},t)) \odot \boldsymbol{z}_k^{\ell,1}(\boldsymbol{x},t) \tag{14}$$

$$\boldsymbol{z}_k^{\ell,1}(\boldsymbol{x},t) = \frac{1}{\sqrt{N}} \boldsymbol{W}_k^{\ell,2}(t)^\top \boldsymbol{g}^{\ell+1}(\boldsymbol{x},t) \tag{15}$$

$$= \underbrace{\frac{1}{\sqrt{N}} \boldsymbol{W}_k^{\ell,2}(0)^\top \boldsymbol{g}^{\ell+1}(\boldsymbol{x},t)}_{\boldsymbol{\xi}_k^{\ell,1}(\boldsymbol{x},t)} + \gamma_0 \mathbb{E}_{\boldsymbol{x}'} \int dt' \Delta(\boldsymbol{x}',t') C_g^{\ell+1}(\boldsymbol{x},\boldsymbol{x}',t,t') \phi(\boldsymbol{h}_k^{\ell,1}(\boldsymbol{x}',t')) \tag{16}$$

We expand out the dynamics

$$p_k^\ell(\boldsymbol{x},t) = \gamma_0 \mathbb{E}_{\boldsymbol{x}'} \int dt' \Delta(\boldsymbol{x}',t') \mathcal{A}_k^\ell(\boldsymbol{x}',t') \dot{\sigma}_k(\boldsymbol{x}',t') C_h^\ell(\boldsymbol{x},\boldsymbol{x}',t,t')$$

$$\mathcal{A}_k^\ell(\boldsymbol{x},t) = \frac{1}{\sqrt{N N_e}} \boldsymbol{g}^{\ell+1}(\boldsymbol{x},t)^\top \boldsymbol{W}_k^{\ell,2}(0) \phi(\boldsymbol{\eta}_k^\ell(\boldsymbol{x},t)) + \gamma_0 \mathbb{E}_{\boldsymbol{x}'} \int dt' \sigma_k(\boldsymbol{x}',t') C_g^{\ell+1}(\boldsymbol{x},\boldsymbol{x}',t,t') C_{\phi_k^1}^\ell(\boldsymbol{x},\boldsymbol{x}',t,t')$$

$$= \underbrace{\frac{1}{N_e} \boldsymbol{\xi}_k^{\ell,1}(\boldsymbol{x},t) \cdot \phi(\boldsymbol{h}_k^{\ell,1}(\boldsymbol{x},t))}_{C_{\xi_k}^\ell} + \gamma_0 \mathbb{E}_{\boldsymbol{x}'} \int dt' \sigma_k(\boldsymbol{x}',t') C_g^{\ell+1}(\boldsymbol{x},\boldsymbol{x}',t,t') C_{\phi_k^1}^\ell(\boldsymbol{x},\boldsymbol{x}',t,t') \tag{17}$$

The only random variables that depend on the random weights that we need to characterize are

$$\boldsymbol{\chi}_k^{\ell,1}(\boldsymbol{x},t) = \frac{1}{\sqrt{N}} \boldsymbol{W}_k^{\ell,1}(0) \boldsymbol{h}^\ell(\boldsymbol{x},t) \ , \ \ \bar{\boldsymbol{\chi}}^{\ell+1}(\boldsymbol{x},t) = \frac{\sqrt{N}}{N_e E} \sum_{k=1}^E \sigma_k(\boldsymbol{x},t) \boldsymbol{W}_k^{\ell,2}(0) \phi(\boldsymbol{h}_k^{\ell,1}(\boldsymbol{x},t)) \tag{18}$$

$$\boldsymbol{\xi}_k^{\ell,1}(\boldsymbol{x},t) = \frac{1}{\sqrt{N}} \boldsymbol{W}_k^{\ell,2}(0)^\top \boldsymbol{g}^{\ell+1}(\boldsymbol{x},t) \ , \ \ \bar{\boldsymbol{\xi}}^\ell(\boldsymbol{x},t) = \frac{\sqrt{N}}{N_e E} \sum_{k=1}^E \sigma_k(\boldsymbol{x},t) \boldsymbol{W}_k^{\ell,1}(0)^\top \boldsymbol{g}_k^{\ell,1}(\boldsymbol{x},t) \tag{19}$$

The global order parameters we will track include the following correlation kernels $C$, response functions

$$C_h^\ell = \frac{1}{N} \boldsymbol{h}^\ell(\boldsymbol{x},t) \cdot \boldsymbol{h}^\ell(\boldsymbol{x}',t') \ , \ \ C_g^\ell(\boldsymbol{x},\boldsymbol{x}',t,t') = \frac{1}{N} \boldsymbol{g}^\ell(\boldsymbol{x},t) \cdot \boldsymbol{g}^\ell(\boldsymbol{x}',t') \tag{20}$$

$$R_{h\xi}^L(\boldsymbol{x},\boldsymbol{x}',t,t') = -\frac{i}{N} \boldsymbol{h}^L(\boldsymbol{x},t) \cdot \widehat{\boldsymbol{\xi}}^L(\boldsymbol{x}',t') \tag{21}$$

$$M_{\sigma\sigma C_\phi}^\ell = \frac{1}{E} \sum_k \sigma_k^\ell(\boldsymbol{x},t) \sigma_k^\ell(\boldsymbol{x}',t') C_{\phi_k}^\ell(\boldsymbol{x},\boldsymbol{x}',t,t') \tag{22}$$

$$M_{\sigma\sigma C_g}^\ell = \frac{1}{E} \sum_k \sum_k \sigma_k^\ell(\boldsymbol{x},t) \sigma_k^\ell(\boldsymbol{x}',t') C_{g_k}^\ell(\boldsymbol{x},\boldsymbol{x}',t,t') \tag{23}$$

$$M_{\sigma\sigma\mathcal{A}\mathcal{A}}^\ell = \frac{1}{E} \sum_k \sigma_k^\ell(\boldsymbol{x},t) \sigma_k^\ell(\boldsymbol{x}',t') \mathcal{A}_k^\ell(\boldsymbol{x},t) \mathcal{A}_k^\ell(\boldsymbol{x}',t') \tag{24}$$

$$\bar{R}_{\phi\xi}^\ell = -\frac{i}{E} \sum_k \sigma_k(\boldsymbol{x},t) \left( \frac{1}{N_e} \phi(\boldsymbol{h}_k^{\ell,1}(\boldsymbol{x},t)) \cdot \widehat{\boldsymbol{\xi}}_k^{\ell,1}(\boldsymbol{x}',t') \right) \tag{25}$$

$$\bar{R}_{g\chi}^\ell = -\frac{i}{E} \sum_k \sigma_k(\boldsymbol{x},t) \left( \frac{1}{N_e} \boldsymbol{g}_k^{\ell,1}(\boldsymbol{x},t) \cdot \widehat{\boldsymbol{\chi}}_k^{\ell,1}(\boldsymbol{x}',t') \right) \tag{26}$$

For each of these variables, there is a corresponding conjugate order parameter. Averaging over the initial random weights for each layer generates the following DMFT path integral over a set of order parameters $\boldsymbol{Q}_{res}$

$$Z = \int d\boldsymbol{Q}_{\text{res}} \exp\left(N\mathcal{S}(\boldsymbol{Q}_{\text{res}})\right) \tag{27}$$

resulting in the following $\mathcal{O}(1)$ action $\mathcal{S}$ where we suppress data and time indices, where we let $\nu = E/N$ and $\alpha_\star = \frac{N}{N_e EL}$

$$\mathcal{S} = \widehat{f}\left(f - \Phi^L \Delta - R_{h\xi}^L\right) - \widehat{R}_{h\xi}^L R_{h\xi}^L + \sum_\ell \left[ C_h^\ell \widehat{C}_h^\ell + C_g^\ell \widehat{C}_g^\ell \right]$$

$$+ \nu \sum_\ell \left[ \widehat{M}_{\sigma\sigma C_\phi}^\ell M_{\sigma\sigma C_\phi}^\ell + \widehat{M}_{\sigma\sigma C_g}^\ell M_{\sigma\sigma C_g}^\ell + \widehat{M}_{\dot\sigma\dot\sigma\mathcal{A}\mathcal{A}}^\ell M_{\dot\sigma\dot\sigma\mathcal{A}\mathcal{A}}^\ell \right] - \frac{1}{\alpha_\star L} \sum_\ell \left[ N_e \widehat{\bar{R}}_{\phi\xi}^\ell \bar{R}_{\phi\xi}^\ell + N_e \widehat{\bar{R}}_{g\chi}^\ell \bar{R}_{g\chi}^\ell \right]$$

$$+ \ln \mathscr{Z}_{\text{res}} + \nu \sum_{\ell=1}^L \ln \mathscr{Z}_{\text{exp}}^\ell$$

$$\nu = \frac{E}{N} \ , \ \alpha_\star = \frac{N}{N_e EL} \tag{28}$$

where the residual stream single site measure is defined as

$$\mathcal{Z}_{\text{res}} = \int \prod_\ell \mathcal{D}\bar{\chi}^\ell \mathcal{D}\widehat{\bar{\chi}}^\ell \mathcal{D}\bar{\xi}^\ell \mathcal{D}\widehat{\bar{\chi}}^\ell \mathcal{D}h^\ell \mathcal{D}\widehat{h}^\ell \mathcal{D}g^\ell \mathcal{D}\widehat{g}^\ell \exp\left(-\frac{\alpha_\star L}{2}\sum_\ell \left[\widehat{\bar{\chi}}^\ell \widehat{\bar{\chi}}^\ell M^\ell_{\sigma\sigma C_\phi} + \widehat{\bar{\xi}}^\ell \widehat{\bar{\xi}}^\ell M^\ell_{\sigma\sigma C_g}\right]\right)$$

$$\exp\left(-i\widehat{R}^L_{h\xi}h^L\xi^L + i\sum_\ell \widehat{\bar{\chi}}^\ell \left[\bar{\chi}^\ell - \bar{R}^\ell_{\phi\xi}g^\ell\right] + i\sum_\ell \widehat{\bar{\xi}}^\ell \left[\bar{\xi}^\ell - \bar{R}^\ell_{g\chi}h^\ell\right]\right)$$

$$\exp\left(i\sum_\ell \widehat{h}^{\ell+1}\left[h^{\ell+1} - h^\ell - L^{-1}\bar{\chi}^\ell - \gamma_0 L^{-1}\Delta M^\ell_{\sigma\sigma C_\phi}g^{\ell+1}\right]\right)$$

$$\exp\left(i\sum_\ell \widehat{g}^\ell \left[g^\ell - g^{\ell+1} - L^{-1}\bar{\xi}^\ell - \gamma_0 L^{-1}\Delta\left(M^\ell_{\sigma\sigma C_g} + M^\ell_{\dot{\sigma}\dot{\sigma}\mathcal{A}\mathcal{A}}\right)h^\ell\right]\right) \tag{29}$$

$$\alpha_\star \equiv \frac{N}{N_e E L} \tag{30}$$

Similarly, the expert moment generating functions $\mathcal{Z}^\ell_{\text{exp}}$ have the form

$$\mathcal{Z}^\ell_{\text{exp}} = \int \mathcal{D}p^\ell \mathcal{D}\widehat{p}^\ell \mathcal{D}\mathcal{A}^\ell \mathcal{D}\widehat{\mathcal{A}}^\ell \mathcal{D}C_{\phi_k}\mathcal{D}\widehat{C}_{\phi_k}\mathcal{D}C^\ell_{g_k}\mathcal{D}\widehat{C}_{g_k}\mathcal{D}C^\ell_{h_k\xi_k}\mathcal{D}\widehat{C}_{h_k\xi_k}$$

$$\exp\left(-\widehat{M}^\ell_{\sigma\sigma C_\phi}\sigma^\ell \sigma^\ell C^\ell_{\phi_k} - \widehat{M}^\ell_{\sigma\sigma C_g}\sigma^\ell \sigma^\ell C^\ell_{g_k} - \widehat{M}^\ell_{\dot{\sigma}\dot{\sigma}\mathcal{A}\mathcal{A}}\dot{\sigma}^\ell \dot{\sigma}^\ell \mathcal{A}^\ell \mathcal{A}^\ell\right)$$

$$\exp\left(i\widehat{p}^\ell_k \left[p^\ell_k - \gamma_0 \Delta\dot{\sigma}^\ell_k \mathcal{A}^\ell_k C^\ell_h\right] + i\widehat{\mathcal{A}}^\ell_k \left[\mathcal{A}^\ell_k - C^\ell_{h_k\xi_k} - \gamma_0 \Delta\sigma^\ell_k C^{\ell+1}_g C^\ell_{\phi_k}\right]\right)$$

$$\exp\left(C_{\phi_k}\widehat{C}_{\phi_k} + C_{g_k}\widehat{C}_{g_k} + C_{h_k\xi_k}\widehat{C}_{h_k\xi_k} + N_e \ln \mathcal{Z}^\ell_{\text{within-exp}}\right) \tag{31}$$

The within-expert distribution is defined as

$$\mathcal{Z}^\ell_{\text{within-exp}} = \int \mathcal{D}\chi^\ell_k \mathcal{D}\widehat{\chi}^\ell_k \mathcal{D}\xi^\ell_k \mathcal{D}\xi^\ell_k \mathcal{D}\widehat{h}^\ell_k \mathcal{D}h^\ell_k \mathcal{D}\widehat{g}^\ell_k \mathcal{D}g^\ell_k \exp\left(-\frac{1}{2}\left[\widehat{\chi}^\ell_k \widehat{\chi}^\ell_k C^\ell_h + \widehat{\xi}^\ell_k \widehat{\xi}^\ell_k C^{\ell+1}_g\right] + i\widehat{\chi}^\ell_k \chi^\ell_k + i\widehat{\xi}^\ell_k \xi^\ell_k\right)$$

$$\exp\left(i\widehat{h}^\ell_k \left(h^\ell_k - \chi^\ell_k - \gamma_0 \Delta\sigma_k C^\ell_h g^\ell_k\right) + i\widehat{g}^\ell_k \left(g^\ell_k - \dot{\phi}(h^\ell_k)\left[\xi^\ell_k + \gamma_0 \Delta\sigma^\ell_k C^{\ell+1}_g \phi(h^\ell_k)\right]\right)\right)$$

$$\exp\left(-\frac{1}{N_e}\widehat{C}^\ell_{\phi_k}\phi(h^\ell_k)\phi(h^\ell_k) - \frac{1}{N_e}\widehat{C}^\ell_{g_k}g^\ell_k g^\ell_k - \frac{1}{N_e}\widehat{C}^\ell_{\phi_k\xi_k}\phi(h^\ell_k)\xi^\ell_k\right)$$

$$\exp\left(-i\widehat{\bar{R}}^\ell_{\phi\xi}\sigma^\ell_k \phi(h^\ell_k)\widehat{\xi}^\ell_k - i\widehat{\bar{R}}^\ell_{g\chi}\sigma^\ell_k g^\ell_k \widehat{\chi}^\ell_k\right) \tag{32}$$

We let $N_e(N)$ and $E(N)$ be diverging functions for the hidden width and expert size as a function of residual stream width $N$ at any fixed value of depth $L(N)$ (possibly constant or diverging). We assume the following condition to be satisfied

$$\lim_{N\to\infty} \frac{N}{N_e(N)E(N)L(N)} = \alpha_\star = 0 \tag{33}$$

This is satisfied for many common scaling strategies. For instance, if FFN ratio $N_e/N$ is fixed and $E$ also diverges (at any rate) then this condition is satisfied as $\alpha_\star = 0$. Similarly, if $E/N$ is fixed and $N_e$ diverges (at any rate) then $\alpha_\star = 0$. We consider simultaneously diverging depth below.

**Saddle Point Equations**  The order parameters $\boldsymbol{Q}_{\text{res}}$ are computed from a saddle point of $\mathcal{S}$. We let $\langle\rangle$ represent an average over the neuron measure defined by $\mathcal{Z}_{\text{res}}$ and let $[]$ represent an average over the expert measure defined by $\mathcal{Z}_{\text{exp}}$ and lastly let $\{\}$ represent an average over the within-expert neuron distribution. Recalling that $\nu = E/N$ and $\alpha_\star = \frac{N}{N_e E L}$, the saddle

point equations are

$$\frac{\partial S}{\partial \widehat{C}_h^\ell} = C_h^\ell - \langle h^\ell h^\ell \rangle = 0 \tag{34}$$

$$\frac{\partial S}{\partial \widehat{C}_g^\ell} = C_g^\ell - \langle g^\ell g^\ell \rangle = 0 \tag{35}$$

$$\frac{\partial S}{\partial \widehat{M}_{\sigma\sigma C_{\phi_k}}^\ell} = \nu M_{\sigma\sigma C_{\phi_k}}^\ell - \nu \left[ \sigma^\ell \sigma^\ell \{ \phi(h_k^\ell)\phi(h_k^\ell) \} \right] = 0 \tag{36}$$

$$\frac{\partial S}{\partial \widehat{M}_{\sigma\sigma C_{g_k}}^\ell} = \nu M_{\sigma\sigma C_{g_k}}^\ell - \nu \left[ \sigma^\ell \sigma^\ell \{ g_k^\ell g_k^\ell \} \right] = 0 \tag{37}$$

$$\frac{\partial S}{\partial \widehat{M}_{\dot{\sigma}\dot{\sigma}\mathcal{A}\mathcal{A}}^\ell} = \nu M_{\dot{\sigma}\dot{\sigma}\mathcal{A}\mathcal{A}}^\ell - \nu \left[ \dot{\sigma}^\ell \dot{\sigma}^\ell \mathcal{A}^\ell \mathcal{A}^\ell \right] = 0 \tag{38}$$

$$\frac{\partial S}{\partial \widehat{R}_{\phi\xi}^\ell} = -\nu \bar{R}_{\phi\xi}^\ell - i\nu \left[ \sigma^\ell \left\{ \phi(h_k^\ell)\widehat{\xi}_k^\ell \right\} \right] = 0 \tag{39}$$

$$\frac{\partial S}{\partial \widehat{R}_{g\chi}^\ell} = -\nu \bar{R}_{g\chi}^\ell - i\nu \left[ \sigma^\ell \left\{ g_k^\ell \widehat{\chi}_k^\ell \right\} \right] = 0 \tag{40}$$

$$\frac{\partial S}{\partial \bar{R}_{\phi\xi}^\ell} = -\frac{1}{\alpha_\star L} \widehat{R}_{\phi\xi}^\ell - i \left\langle \widehat{\bar{\chi}}^\ell g^\ell \right\rangle = 0 \tag{41}$$

$$\frac{\partial S}{\partial \bar{R}_{g\chi}^\ell} = -\frac{1}{\alpha_\star L} \widehat{R}_{g\chi}^\ell - i \left\langle \widehat{\bar{\xi}}^\ell h^\ell \right\rangle = 0 \tag{42}$$

The remaining saddle point equations give $\widehat{C} = 0$ and $\widehat{M} = 0$. The response functions can be rearranged as derivatives through integration by parts (Crisanti & Sompolinsky, 2018; Mignacco et al., 2020; Bordelon & Pehlevan, 2022; 2026)

$$-i\{\phi(h_k^\ell)\widehat{\xi}_k^\ell\} = \left\{ \frac{\partial \phi(h_k^\ell)}{\partial \xi_k^\ell} \right\} \; , \; -i\{g_k^\ell \widehat{\chi}_k^\ell\} = \left\{ \frac{\partial g_k^\ell}{\partial \chi_k^\ell} \right\}. \tag{43}$$

We note that the functions $\widehat{R}_{\phi\xi}^\ell$ and $\widehat{R}_{g\chi}^\ell$ are both $\mathcal{O}\left( \frac{N}{N_e E L} \right) = \mathcal{O}(\alpha_\star)^3$. We thus introduce the following rescaled definitions to simplify our expression

$$\widehat{R}_{\phi\xi}^\ell \equiv \alpha_\star A^\ell \; , \; \widehat{R}_{g\chi}^\ell \equiv \alpha_\star B^\ell \tag{44}$$

We will express the final single site equations in terms of $A^\ell$ and $B^\ell$ which are dimensionless at leading order.

**Top-K Operation**    The top-K gating operation for $\kappa = K/E$ is well defined as a quantile thresholding operation under the mean-field measure over experts (averages represented by $[]$). Introduce the random variable $q = \sigma(p) + b$ and let $q_\star(\kappa)$ represent the solution to the equation

$$\left[ 1_{q \geq q_\star(\kappa)} \right] = \kappa \tag{45}$$

The variable $q_\star(\kappa)$ is thus the lower end point of integration for the gating preactivation distribution that captures the top $\kappa$ probability mass. The hard-routing gate variables which occur in the top-K routing are thus

$$\sigma^\ell(x,t) = 1_{q \geq q_\star(\kappa)} \sigma \left( p^\ell(\boldsymbol{x},t) \right) \tag{46}$$

These are the hard gating variables which govern the evolution equations.

---

[3]To see this compute $\frac{\partial h}{\partial r}$ or $\frac{\partial g}{\partial u}$ and see that they are $\sim \mathcal{O}(L^{-1})$. Using the fact that $\frac{1}{\nu N_e} = \frac{N}{N_e E}$, this verifies the correct scale of $\widehat{R}$

**DMFT Equations** The DMFT single site equations under the assumption that

$$\alpha_\star \equiv \frac{N}{N_e LE}, \tag{47}$$

are the following for the the residual stream

$$
\begin{aligned}
h^{\ell+1}(\boldsymbol{x}, t) &= h^\ell(\boldsymbol{x}, t) + L^{-1} u^\ell(\boldsymbol{x}, t) \\
&\quad + L^{-1} \int dt' d\boldsymbol{x}' \left[ \bar{R}^\ell_{\phi\xi}(\boldsymbol{x}, \boldsymbol{x}', t, t') + \gamma_0 p(\boldsymbol{x}') \Delta(\boldsymbol{x}', t') M^\ell_{\sigma\sigma C_{\phi_k}}(\boldsymbol{x}, \boldsymbol{x}', t, t') \right] g^{\ell+1}(\boldsymbol{x}', t') \\
g^\ell(\boldsymbol{x}, t) &= g^{\ell+1}(\boldsymbol{x}, t) + L^{-1} r^\ell(\boldsymbol{x}, t) \\
&\quad + L^{-1} \int dt' d\boldsymbol{x}' \left[ \bar{R}^\ell_{g\chi}(\boldsymbol{x}, \boldsymbol{x}', t, t') + \gamma_0 p(\boldsymbol{x}') \Delta(\boldsymbol{x}', t') M^\ell_{\sigma\sigma C_{g_k}}(\boldsymbol{x}, \boldsymbol{x}', t, t') \right] h^\ell(\boldsymbol{x}', t') \\
u^\ell(\boldsymbol{x}, t) &\sim \mathcal{GP}\left( 0, \alpha_\star L M^\ell_{\sigma\sigma C_{\phi_k}}(\boldsymbol{x}, \boldsymbol{x}', t, t') \right), \quad r^\ell(\boldsymbol{x}, t) \sim \mathcal{GP}\left( 0, \alpha_\star L M^\ell_{\sigma\sigma C_{g_k}}(\boldsymbol{x}, \boldsymbol{x}', t, t') \right)
\end{aligned} \tag{48}
$$

All averages $\langle \rangle$ computed from the residual stream are averages over the above stochastic processes. We note that the variable $\alpha_\star$ only appears in the variance of the stochastic process $u^\ell$ and $r^\ell$ on the forward and backward passes respectively.

Next, for the expert distribution, we have the following DMFT equations for router preactivation $p$

$$
\begin{aligned}
p^\ell(\boldsymbol{x}, t) &= \gamma_0 \int dt' \mathcal{A}^\ell(\boldsymbol{x}', t') \dot{\sigma}^\ell(\boldsymbol{x}', t') H^\ell(\boldsymbol{x}, \boldsymbol{x}', t, t') \\
b(t) &= b(0) + \gamma_0 \int dt' \left( \kappa - \mathbb{E}_{\boldsymbol{x}} 1_{q^\ell(\boldsymbol{x}, t) \geq q^\ell_\star(\boldsymbol{x}, t)} \right) \\
\mathcal{A}^\ell(\boldsymbol{x}, t) &= C^\ell_{\phi_k \xi_k}(\boldsymbol{x}, \boldsymbol{x}, t, t) + \gamma_0 \mathbb{E}_{\boldsymbol{x}'} \int dt' \sigma^\ell(\boldsymbol{x}', t') C^{\ell+1}_g(\boldsymbol{x}, \boldsymbol{x}', t, t') C^\ell_\phi(\boldsymbol{x}, \boldsymbol{x}', t, t') \\
\sigma^\ell(\boldsymbol{x}, t) &= 1_{q^\ell(\boldsymbol{x}, t) \geq q^\ell_\star(\boldsymbol{x}, t)} \, \sigma(p^\ell(\boldsymbol{x}, t))
\end{aligned} \tag{49}
$$

These equations define the averaging operation over experts $[]$. The main source of disorder from the initial condition arises from the random initial biases $b(0)$ for the router. Lastly the mean field dynamics of each neuron within the experts have the form

$$h^\ell_k(\boldsymbol{x}, t) = u^\ell_k(\boldsymbol{x}, t) + \alpha_\star \int dt d\boldsymbol{x}' A^\ell(\boldsymbol{x}, \boldsymbol{x}', t, t') g^\ell_k(\boldsymbol{x}', t') \tag{50}$$

$$+ \gamma_0 \mathbb{E}_{\boldsymbol{x}'} \int dt' \Delta(\boldsymbol{x}', t') \sigma^\ell_k(\boldsymbol{x}', t') C^\ell_h(\boldsymbol{x}, \boldsymbol{x}', t, t') g^\ell_k(\boldsymbol{x}', t'), \; u^\ell_k(\boldsymbol{x}, t) \sim \mathcal{GP}(0, C^\ell_h)$$

$$z^\ell_k(\boldsymbol{x}, t) = r^\ell_k(\boldsymbol{x}, t) + \alpha_\star \int dt d\boldsymbol{x}' B^{\ell+1}(\boldsymbol{x}, \boldsymbol{x}', t, t') \phi(h^\ell_k(\boldsymbol{x}', t')) \tag{51}$$

$$+ \gamma_0 \mathbb{E}_{\boldsymbol{x}'} \int dt' \Delta(\boldsymbol{x}', t') \sigma^\ell_k(\boldsymbol{x}', t') C^{\ell+1}_g(\boldsymbol{x}, \boldsymbol{x}', t, t') \phi(h^\ell_k(\boldsymbol{x}', t')), \; \xi^\ell_k(\boldsymbol{x}, t) \sim \mathcal{GP}(0, C^{\ell+1}_g)$$

$$g^\ell_k(\boldsymbol{x}, t) = \dot{\phi}(h^\ell_k(\boldsymbol{x}, t)) z^\ell_k(\boldsymbol{x}, t) \tag{52}$$

The average $\{\}$ over neurons within each expert are determined by the above stochastic process. We see that the response terms involving $A^\ell$ and $B^\ell$

**Large Depth Limit**   The large depth limit simply introduces a differential equation in layer time $\tau = \ell/L \in [0,1]$ (Bordelon et al., 2023; Dey et al., 2025). The order parameters become functions of this "depth-time" $\tau$

$$
\begin{aligned}
C_h^\ell(\boldsymbol{x}, \boldsymbol{x}', t, t')|_{\ell=\lfloor \tau L \rfloor} &\to C_h(\tau, \boldsymbol{x}, \boldsymbol{x}', t, t') \\
C_g^\ell(\boldsymbol{x}, \boldsymbol{x}', t, t')|_{\ell=\lfloor \tau L \rfloor} &\to C_g(\tau, \boldsymbol{x}, \boldsymbol{x}', t, t') \\
M_{\sigma\sigma C_\phi}^\ell(\boldsymbol{x}, \boldsymbol{x}', t, t')|_{\ell=\lfloor \tau L \rfloor} &\to M_{\sigma\sigma C_\phi}(\tau, \boldsymbol{x}, \boldsymbol{x}', t, t') \\
M_{\sigma\sigma C_g}^\ell(\boldsymbol{x}, \boldsymbol{x}', t, t')|_{\ell=\lfloor \tau L \rfloor} &\to M_{\sigma\sigma C_g}(\tau, \boldsymbol{x}, \boldsymbol{x}', t, t') \\
M_{\dot\sigma\dot\sigma \mathcal{A}\mathcal{A}}^\ell(\boldsymbol{x}, \boldsymbol{x}', t, t')|_{\ell=\lfloor \tau L \rfloor} &\to M_{\dot\sigma\dot\sigma \mathcal{A}\mathcal{A}}(\tau, \boldsymbol{x}, \boldsymbol{x}', t, t') \\
\bar{R}_{\phi\xi}^\ell(\boldsymbol{x}, \boldsymbol{x}', t, t')|_{\ell=\lfloor \tau L \rfloor} &\to \bar{R}_{\phi\xi}(\tau, \boldsymbol{x}, \boldsymbol{x}', t, t') \\
\bar{R}_{g\chi}^\ell(\boldsymbol{x}, \boldsymbol{x}', t, t')|_{\ell=\lfloor \tau L \rfloor} &\to \bar{R}_{g\chi}(\tau, \boldsymbol{x}, \boldsymbol{x}', t, t')
\end{aligned}
\tag{53}
$$

We further assume that $N, L, N_e$ are all diverging functions of residual steram width $N$

$$
\alpha_\star \equiv \lim_{N \to \infty} \frac{N}{L(N)E(N)N_e(N)}
\tag{54}
$$

Similarly, we have an SDE description for the residual stream variables which are driven by an additional Brownian motion $du(\tau, \boldsymbol{x}, t)$

$$
dh(\tau, \boldsymbol{x}, t) = \sqrt{\alpha_\star} du(\tau, \boldsymbol{x}, t) + d\tau \int dt' d\boldsymbol{x}' \left[ \bar{R}_{\phi\xi}^\ell(\boldsymbol{x}, \boldsymbol{x}', t, t') + \gamma_0 p(\boldsymbol{x}')\Delta(\boldsymbol{x}', t') M_{\sigma\sigma C_{\phi_k}}^\ell(\boldsymbol{x}, \boldsymbol{x}', t, t') \right] g(\tau, \boldsymbol{x}', t')
$$
$$
\langle du(\tau, \boldsymbol{x}, t) du(\tau', \boldsymbol{x}', t') \rangle = d\tau d\tau' \delta(\tau - \tau') M_{\sigma\sigma C_\phi}(\tau, \boldsymbol{x}, \boldsymbol{x}', t, t').
\tag{55}
$$

For $\alpha_\star = 0$, the Brownian motion term disappears and the residual stream dynamics reduce to a neural ODE, consistent with CompleteP scaling (Dey et al., 2025; Chizat, 2025). Further, for $\alpha_\star = 0$ the response function terms for the within-FFN block neuron dynamics disappear, resulting in simpler within block dynamics. This analysis points to finite $\alpha_\star$ as worthy of future study as they provide an example of complete feature learning in a Neural SDE, which was previously thought to require an ODE limit (Bordelon et al., 2024; Dey et al., 2025).

**Key Finding: FFN Ratio Indepedence**   The ratio of the FFN width $N_e/N$ only appears in the ratio $\alpha_\star$. Provided this variable converges rapidly to zero as the model approaches the limit, the limit dynamics will not depend on the FFN ratio. In practice, most of the common joint scaling strategies scale depth and expert width linearly with the residual width, which results in $\alpha_\star \sim \frac{1}{N} \to 0$.

### E.1. Error Analysis

In this section, we discuss the asymptotic errors to the joint scaling limit. We are interested in quantifying the scale of the error in network outputs $f$ at finite $N, E, N_e, L$ and the joint infinite limit with $\alpha_\star = \frac{N}{EN_eL} \to 0$. Concretely, we aim to bound

$$
|f(N, E, N_e, L) - f_\infty|.
\tag{56}
$$

The key sources of finite size errors near the asymptotic limit are

1. **Finite Residual Stream Fluctuations**: Imperfect concentration of residual stream kernels $C_h$ and $C_g$ induce central limit theorem fluctuations on scale $\mathcal{O}(N^{-1/2})$. These fluctuations also impact the model outputs.

2. **Finite Expert Fluctuations** Imperfect concentration of expert averaged kernels $M_{\sigma\sigma C_\phi}, M_{\sigma\sigma C_g}$, etc. induce errors on the order of $\mathcal{O}(E^{-1/2})$.

3. **Finite Expert Size Corrections** The finite expert size effects contribute in two places. First, they impact the residual stream dynamics at leading order at a scale $\mathcal{O}(E^{-1/2}N_e^{-1/2} + N_e^{-1})$ as they perturb $M$ kernels. However, since these kernels are also expert averages, the CLT fluctuations scale as for large $E$ (mean-zero fluctuations from finite $N_e$ in the expert averages {} average out to zero when we compute averages over the expert population []). Second, the within-expert response functions at finite $\alpha_\star$ perturb the within-expert dynamics by a scale $\alpha_\star = \frac{N}{N_eEL}$.

4. **Noise on Residual Stream vs Neural ODE** The contribution from the $u^\ell$ and $r^\ell$ variables contribute variance on the scale $\alpha_\star = \frac{N}{LN_eE}$. These cause an error in the predictor at a scale of $\sim \alpha_\star$. We note that our error metric is focused on the error in the *kernels and model outputs, not on the residual stream itself* which would have $\sqrt{\alpha_\star}$ as the error.

5. **Finite Depth Discretization** The kernels at large depth $L$ can be viewed as an Euler discretization $C_h^{\ell+1} = C_h^\ell + \frac{1}{L}[...]$ of an underlying ODE. This discretization induces an error of order $\sim 1/L$.

In total we arrive at an error bound of the form

$$|f(N, E, N_e, L) - f_\infty| = \mathcal{O}\left(\frac{1}{\sqrt{N}} + \frac{1}{\sqrt{E}} + \frac{1}{N_e} + \frac{N}{N_eEL} + \frac{1}{L}\right). \tag{57}$$

The approximation error is minimized by taking the following joint scaling as the residual stream width $N$ diverges

$$E \sim N \,,\ L \sim \sqrt{N} \,,\ N_e \sim \sqrt{N} \tag{58}$$

Under this regime $\alpha_\star \sim \frac{1}{N} \to 0$ as desired. If we define the total parameters $P = NEN_eL \sim N^3$ then the scaling law for the error should be $\mathcal{O}(P^{-1/6})$.

### E.2. Verifying Convergence in Soft Routing Model

In Figure 19 we verify the convergence of the dynamics to the theoretical mean field limit. The model is trained on a multi-index model of degree 4 in $D = 20$ dimensions. We do not apply any hard sparsity gating in these simulations in order to provide a minimal model that displays the correct scaling behaviors. We plot the loss dynamics and the empirical $p_k(t)$ measures which are consistent across $E, N, N_e \gg 1$.

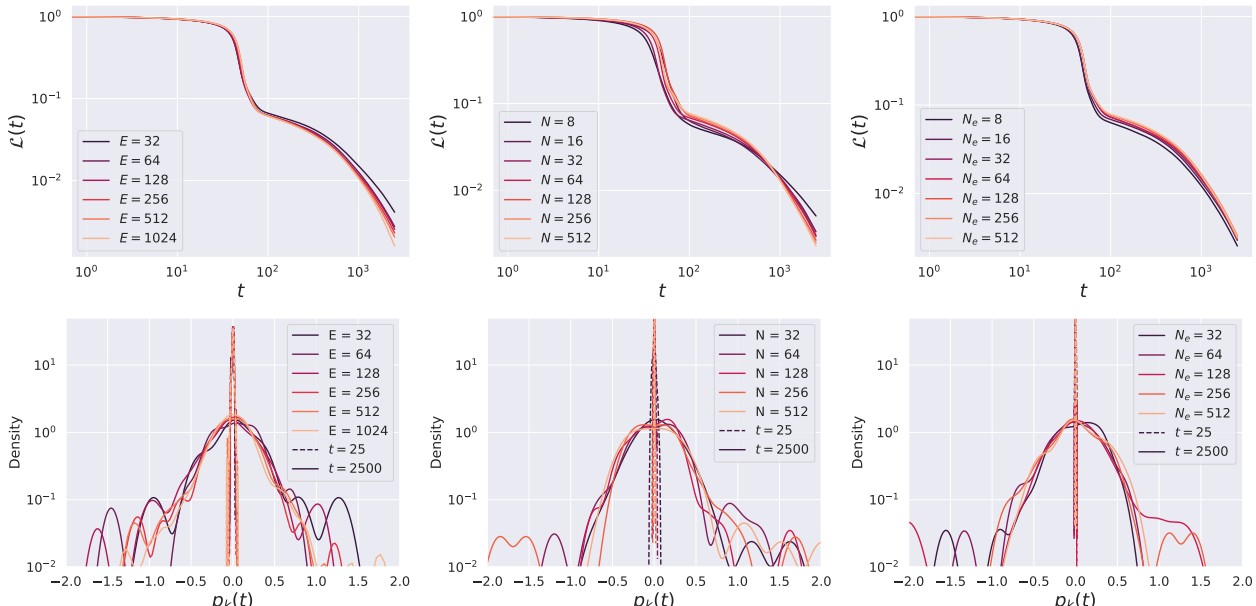

*Figure 19.* Loss dynamics $\mathcal{L}(t)$ and router preactivation $p_k(t)$ distribution (over experts) in a soft router MLP model trained on a multi-index polynomial target function. The parameterization achieves convergence to a well defined limiting dynamical system provided $\frac{N}{N_eE}$ is finite. From left to right: constant $N$ and $N_e$ (left), constant FFN ratio $N_e/N$ with increasing $N$ (middle), and constant $N$ with increasing $N_e$.

