# OpenReview forum: "Hyperparameter Transfer with Mixture-of-Expert Layers"
_ICML.cc/2026/Conference — ICML 2026 regular_

### Official Review · Reviewer_t9fY · 2026-03-10

**Soundness:** 3
**Presentation:** 3
**Significance:** 3
**Originality:** 3
**Overall Recommendation:** 5
**Confidence:** 3

**Summary:**

In this work, the authors' propose a hyperparameter (HP) choice scheme for mixture-of-experts (MoE) architectures as they scale in width, depth, number of experts, and expert width. These schemes are given in Table 1. They follow in the line of work established by muP (Yang and Hu 2022) and completeP (Dey et al. 2025) that provide guidance for the HP choices as the model scales in width and depth, respectively. The underlying idea is, with the hyperparameters at certain scales, the network's training dynamics approach a deterministic limit independent of the network's size and dependent only on hyperparameters. However, those scales vary across architectural components and must be chosen for each component. In this work, the authors derive the HP choice schemes for the MoE Router, expert load-balancing bias, and individual expert MLPs. They justify their scheme using arguments from Dynamic Mean Field Theory (DMFT).  The authors show experiments while training MoE architectures on the FineWeb and C4 datasets showing that their scheme indeed enable HP transfer. The authors also note auxiliary take away from their theory (supported by experiments) such as the fact that increasing number of experts is more parameter efficient than increasing expert size (Figure 5).

**Compliance With Llm Reviewing Policy:**

Affirmed.

**Final Justification:**

In this work, the authors' propose a hyperparameter (HP) choice scheme for mixture-of-experts (MoE) architectures as they scale in width, depth, number of experts, and expert width. These schemes are given in Table 1. They follow in the line of work established by muP (Yang and Hu 2022) and completeP (Dey et al. 2025) that provide guidance for the HP choices as the model scales in width and depth, respectively. The underlying idea is, with the hyperparameters at certain scales, the network's training dynamics approach a deterministic limit independent of the network's size and dependent only on hyperparameters. However, those scales vary across architectural components and must be chosen for each component. In this work, the authors derive the HP choice schemes for the MoE Router, expert load-balancing bias, and individual expert MLPs. They justify their scheme using arguments from Dynamic Mean Field Theory (DMFT).  The authors show experiments while training MoE architectures on the FineWeb and C4 datasets showing that their scheme indeed enable HP transfer. The authors also note auxiliary take away from their theory (supported by experiments) such as the fact that increasing number of experts is more parameter efficient than increasing expert size (Figure 5).

In their rebuttal, the authors further promised to give a more careful account of the scenarios in which their theory would and would not apply: I believe this will be a good help to practioners.

I think this is a scholarly work and recommend acceptance.

**Key Questions For Authors:**

The assumptions and limitations of the theory are scatter throughout the paper and the appendix. This makes it rather difficult to judge whether a given architecture might benefit from the authors' proposed HP choice method. Might the authors give a concise list of (1) what the assumptions and limitations are for the existing theory, (2) what criteria an architecture should satisfy for it to credibly benefit from this proposed scheme, and (3) what could be "no go" architectural choices that would disqualify an architecture from benefitting from this HP choice heuristic?

While the answers to some of these are scattered across the paper and appendix in various places, it would help readers who wish to use this paper as a "how-to" guide if they were instead encapsulated into modular sections or lists.

**Limitations:**

Somewhat. The assumptions and limitations of the theory are scatter throughout the paper and the appendix.

**Strengths And Weaknesses:**

Strengths: This paper provides a simple-to-implement recommendation for hyperparameter choice (and transfer) in MoE architectures. The authors justify their scheme using DMFT and provide experiments demonstrating the practicality of their methods.

Weaknesses: The HP choice scheme is justified by DMFT in very specific architecture realizations. The theory itself require certain scalings of architecture layer outputs. Moreover, the theory is technically only for for MoE architectures with MLP experts and no attentions layers. Thus, it is not easy to see when the proposed HP choice scheme applies and when assumption violations might cause it to fail.

---

> ### Author Rebuttal · Authors · 2026-03-27
>
> We deeply appreciate the insightful comments. In the next version (with the added page limit), we will incorporate the feedback into a detailed conclusionary section. Below we address technical comments:
>
> **1. Architecture is specific and doesn’t necessarily generalize.**
>
> Thank you for pointing this out. Architectural generalizability is an important point that we will certainly clarify in our revision. We already try to maximize our generality by exploring the scaling capability of multiple, if not all, practically relevant model dimensions (e.g. as opposed to fixing all others and only scale width). Architecture-wise, there still exist some flexibilities that can be adopted without major barriers, e.g. if we replace sigmoid expert weighting to softmax gates, or if we stack vertically or parallel additional dense blocks in the residual stream.
> We also want to stress that our major goal in this work is, instead of covering all possible MoE architectures in general, to deliberately choose **one** set of design choices (which happen to be popular amongst industrial frontier open source LLMs) and theoretically justify a set of scaling rules which also empirically shows (1) reliable scaling with HP transfer and (2) performant behavior with minimal tuning.
>
> **2. What criteria an architecture should satisfy to benefit from our analysis, and what architectural choices will be disqualifying?**
>
> Thank you for raising this very interesting question. While there some architectural twists are likely to follow from existing analysis already, it is highly non-trivial what a general set of sufficiency rules will be for an architecture to converge to a consistent and unique mean-field limit. One exemplary necessary condition architecture-wise is that each block needs to have an asymptotic mean-field structure. For instance, in the treatment of (a growing number of) experts, we need to study the mean-field distribution over experts. As a result, fixing a top one activated expert amongst a growing number of total experts doesn’t fit in our analysis perfectly, whereas a well-defined limit exists for fixed sparsity (i.e. the set of activated experts represent conditioning on an event with positive probability). Further necessary conditions on the optimization process also apply, see below.
>
> **3. Assumptions and limitations for the theory?**
>
> Thank you for the comment and we will make sure to include a dedicated discussion on assumptions and limitations in the next version. Certainly, the aforementioned limitations on architecture is one of them. Conditioning on our architecture specifically, the major assumptions/limitations in our existing theory are:
> (1) scale-matched population Grad flow/descent will align with Adam’s scaling behavior with finite batch size, and
> (2) mean-field asymptotic convergence to scaling limit and limiting behaviors can be observed realistically (with moderately large yet finite widths, depth, expert counts, and expert sizes).
> For both limitations, our experiments show that theory remains faithful (finite batch, practical training horizon, and finite model dimension) and hence useful in practice. We leave a more careful treatment of theoretically analyzing Adam with finite batch size and non-asymptotics to future work.
>
> **4. DMFT is only for MoE-FFN with no attention layers.**
>
> Thank you for pointing this out. Existing work on DMFT derives self-attention scaling limits fixing head-dim (empirically popular choice in e.g. Flash-Attn) as well as HP transfer. In our DMFT analysis, since the goal is only to derive and verify insights for the MoE architecture, the simplified model suffices to justify our main goals already. We believe it is straightforward, though a bit tedious, to include the multihead self attention residual blocks because the dynamics of the MHSA blocks will be independent of the MoE block dynamics at infinite width. At finite width, the MHSA dynamics depend by symmetry on MoE blocks only through empirical averages over experts. This basically reduces the analysis of MoE + MHSA blocks to our analysis combined with prior MHSA-only DMFT. We will make sure to clarify this point in the revision.
>
> **We hope the above addresses your concerns. We welcome any remaining questions and feedback.**

---

> > ### Author Rebuttal · Reviewer_t9fY · 2026-04-03
> >
> > The authors have acknowledged my concerns and offered solutions to address them.

---

> > > ### Author Response · Authors · 2026-04-03
> > >
> > > Thank you for your valuable review and feedback comments, which we will adopt to improve the paper in the revision.

---

### Official Review · Reviewer_2AVh · 2026-03-11

**Soundness:** 3
**Presentation:** 3
**Significance:** 3
**Originality:** 3
**Overall Recommendation:** 4
**Confidence:** 3

**Summary:**

This paper addresses the pain point of "difficulty in reliably transferring hyperparameters with scale" in MoE pre-training. It proposes a set of parameterization/scaling rules for decoder-only Transformers with MoE FFNs, enabling hyperparameter transfer of learning rate and initialization variance when simultaneously increasing the width nembd, depth L, number of experts nexp, and expert FFN width multiplier αffn. Experiments primarily use a fixed token budget (approximately 1B tokens; 2000 steps; 500K tokens per step; context=1024; first 1000 steps of warmup) to verify that the "optimal base LR/init" tuned to the base model can be transferred to larger scales on FineWeb (κ=1/4) and C4 (κ=1/12). Furthermore, it demonstrates that performance/stability issues may arise if a constant-level multiplier is not tuned.

**Compliance With Llm Reviewing Policy:**

Affirmed.

**Final Justification:**

This paper is well-written overall, and the authors have made every effort to address the issues in the rebuttal. I am not certain about the paper's assumptions or whether the results can be applied to real-world industrial models; these can be confirmed by other reviewers. From other perspectives, I recommend acceptance.

**Key Questions For Authors:**

Does this still hold true for more standard MoE implementations? Please provide at least one set of transfer results under topK-softmax router + common aux-load-balancing loss (or mainstream open-source implementation), and explain the differences from the current sigmoid + bias-only balance.

What impact would the existence of Share Expert have? Will the same conclusion be reached if the model becomes very fine-grained?

Can the sparsity of other dimensions be derived using the same method, such as the dynamic calculation of hidden dimensions or depth?

**Limitations:**

Yes

**Strengths And Weaknesses:**

Strengths

The topic is important and practical: High cost and instability of HP tuning in MoE training is a real problem in both engineering and research. This paper provides group scaling laws for router/expert up-down/bias.

It offers comprehensive dimensionality coverage: It not only discusses width/depth but also explicitly discusses expert count and expert hidden multiplier, emphasizing the motivation and empirical observations for fixing sparsity κ.

The theoretical motivation is relatively solid: It presents the hierarchical mean field structure and conditions from the perspective of DMFT.

Several "additional experiments" provide support: The appendix mentions the exploration of weight decay, more transfer combinations, the addition of a longer horizon, and the convergence of expert load imbalance with scale.

Weaknesses

The author explicitly acknowledges that each parameter group requires constant-level multiplier tuning, which is equivalent to introducing an additional layer of parameter tuning on top of the base model. The cost of this should be quantified.

The main result depends on the 1B token, and the warmup accounts for half of the step. However, the most valuable scenario for HP transfer is often longer, near-compute-optimal training. The authors also acknowledge in their conclusion that this is still a work for the future.

The paper describes a router using sigmoid, hard top-nact, and an activated set as no-grad, with balance updates relying on bias-only. The question is whether this can generalize to topK-softmax/aux loss MoE?

---

> ### Author Rebuttal · Authors · 2026-03-26
>
> We deeply appreciate the insightful comments. In the next version (with the added page limit), we will incorporate the feedback into a detailed section on assumptions and limitations, theory vs empirics, and future works. Below we address technical comments:
>
> **1. Parameter group constant-level tuning.**
>
> Thank you for bringing this up. Indeed, experimentally we needed to tune constant multipliers in the small (50M) **base model only** to ensure training stability. We want to note that:
>
> 1. Making constant-scale tuning in different types of weights for training stability is a standard and studied practice, even in dense models when training is inherently more stable. Our message is that for MoEs, a minimal level of tuning is not only better for performance but also necessary for stability.
>
> 2. Empirically, one wants to scale up from a small-scale base model (on which tuning is cheap and efficient) that is tested to be both stable and performant, as base model misalignments/suboptimality could likely get exaggerated when scaling up (even if the scaling recipe is correct).
>
> 3. Regarding the quantitative cost, while a grid exhaustive search may still be costly despite in the base model, recent works such as (Mlodozeniec et al, 2025) propose algorithms that exploit optimization landscapes, leading to vast simplifications. In our experiments, we only did simple, computationally un-demanding, coordinate searches on a reduced number of parameter groups, which already suffices for our desired stability. Our results show that **without** heavy tuning for SOTA constants, one can already up-scale reliably and performantly with the correct theoretical recipe, so long as the base model is stable across LR.
>
> **2. Specific MoE architecture, aux-loss, and architecture generality.**
>
> Thank you for pointing this out. Architectural generalizability is an important point that we will certainly clarify in our revision. Sigmoid gating versus softmax is not a critical requirement in our recipe theoretically or empirically. In experiments, we see entirely identical behavior in softmax expert weighting vs sigmoid and there's no evidence that the weighting will be a crucial factor. Our choice of sigmoid routing for presented experiments was due to popularity in modern open-source LLMs (e.g. DeepSeekMoE, DeepSeek V3/R1, Kimi K2.5, Arcee-Trinity). We will further clarify on this point in our revision.
>
> Regarding aux loss, we made this choice deliberately as it is not perfectly compatible with the HP transfer frameworks. An aux. loss (or most of global regularizers) carries with it many complex practitioner-specific desiderata of how one ought to balance the tradeoff of gradients for each parameter group between the primary data-fitting objective and the regularizer (e.g. should layer-1 self-attention weights be equally responsible for load-balancing in layer-10, or should only routers be penalized by their respective layer imbalance independently), as well as how to evaluate models across HPs (e.g. what if some HP improves model loss but slows down balancing).  It is unclear how to do this in a principled and rigorous manner, and it appears that fine-grained, practitioner-specific desiderata will lead to drastically different scaling analyses and training recipes. We discuss this briefly in Apdx D.2 and will further clarify in our revision.
>
> Finally, we stress that the main goal of our work is not to cover all possible MoE implementations (as there appears to be many variants amongst industry) but to study  **one** architecture (popular in frontier open source LLMs by DeepSeek, Kimi, and Arcee) and show that our recipe allows (1) reliably scaling with HP transfer backed by theory; (2) achieves good performance with minimal tuning.
>
> **3. What happens with shared experts / fine-grained experts. Mixture-of-Depth or sparsity in other dim.**
>
> Our result includes fine-grained experts naturally when scaling the number of experts (fixing sparsity). Regarding shared experts, under the normalization where the shared expert is weighted to match the MoE experts, we expect theoretical results on the scaling limit to persist, similar to adding an extra MLP-FFN to the architecture. However, due to imbalanced batches in the backward pass, empirically one almost certainly requires further constant tuning and careful, architecture-specific, gradient handling. We leave it as an interesting open question worth investigating empirically.
>
> We also thank the reviewer for pointing out potential future directions in alternative sparse dimensions. Our theory generalizes to dynamic expert allocation (Expert-Choice like) routing, but MoDs and other sparsities remain open to future work.
>
> **We hope the above addresses your concerns. We welcome any remaining questions and kindly ask that you consider raising your score if they are resolved.**

---

> > ### Author Rebuttal · Reviewer_2AVh · 2026-04-03
> >
> > Thank you for the authors' reply. Their response was very good, but it would be even better if some experimental evidence could be provided. However, regardless, the quality of this article is acceptable.

---

> > > ### Author Response · Authors · 2026-04-03
> > >
> > > Thank you for your valuable review and feedback comments, which we will adopt to improve the paper in the revision. We are committed to adding architecture ablations with softmax in the next version, and we apologize that due to hardware availability, they were not included in our last response in time.
> > >
> > > Here we provide a systematic empirical test of mixing experts with softmax gates, and show that transfer behaviors with sigmoid can be reliably reproduced with softmax. The (anonymous) link is **here** (https://ibb.co/XrWLCTbK).
> > >
> > > We hope they address your concerns and welcome additional questions or feedback. If the reviewer thinks that our response is sufficient, we will also deeply appreciate if you could consider raising the score.

---

### Official Review · Reviewer_BLe8 · 2026-03-15

**Soundness:** 3
**Presentation:** 2
**Significance:** 3
**Originality:** 3
**Overall Recommendation:** 3
**Confidence:** 3

**Summary:**

MoEs introduce new scaling dimensions that have no immediate parameterization rule that ensure stable training dynamics transfer across scales. The authors propose an extension of µP and CompleteP to transformer architectures with MoE layers, deriving scaling rules for initialization standard deviation and learning rate for the MoE parameter groups: router weights, expert biases, and expert up/down projections, expressed as a function of embedding width, depth, expert count, and mlp expansion factor. The sparsity ratio (percentage of active experts) is held fixed across scaling and load balancing is enforced via an auxiliary-loss-free bias update. The parameterization is grounded in an analysis derived from Dynamical Mean Field Theory, showing a three-level mean-field hierarchy over residual stream neurons, expert outputs, and within-expert neurons. This enables the identification of conditions under which training dynamics converge to a scale-invariant fixed point. The authors validate their results on Fineweb and C4 datasets across models ranging from 20M to 1.8B total parameters, sweeping LR and init scales across the scaling dimensions. The authors also observe a competitive pre-training performance against dense baselines, matched for active parameter count.

**Compliance With Llm Reviewing Policy:**

Affirmed.

**Final Justification:**

We appreciate your efforts to restate clearly the assumptions and limitations of your work. The discussion has helped better define the scope of the paper and its positioning relative to prior work. In light of this, we are reassessing our significance score to 3/4.

**Key Questions For Authors:**

Finding 1.2 explicitly excludes depth and Figure 17 reports load balancing instability under depth scaling. Could the authors clarify to which extent the proposed MoE extension interfere with the depth transfer of CompleteP? An appropriate ablation could strengthen the significance of the paper.

The DMFT assumes gradient flow while experiments use Adam. Authors connect both with the SignGD approximation. How tight is this justification in practice? Can we expect settings or regimes where results differ?

The fixed sparsity assumption is shown to be necessary for transfer (Figure 10), yet fixing nact is common in practice for hardware reasons. Is there a way to accommodate the methodology for fixed n_act?

It is unclear why DFMT needs to be used for the parameterization when it didn't seem needed for dense transformers. Why does extending completeP to MoE requires using DMFT and not just MFT?

**Limitations:**

yes

**Strengths And Weaknesses:**

strengths:
The contribution scope is well-defined and addresses a concrete need, given the recent success of MoE and mu-P and the lack of guidance for parameterizing MoE parameter groups. The proposed scaling rules and the required assumptions are clear and the experimental results are conclusive in favor of the proposition.

The experiment set is rich, testing transfer over different scaling dimensions, two datasets, two sparsity levels and running various auxiliary tests and ablations. The ablation of figure 11 depicts the interest of using their method rather than the standard fan-in initialization, which leads to different loss curves across scales, breaking the desired scale in-variance.

The effort of justifying the parameterization using the Dynamical Mean Field Theory sounds promising, though it deviates from my line of work and I find hard to understand the intuition, the necessity and the actual impact of this analysis.

Furthermore, models trained with their scaling framework, transferred from small base models achieve competitive pre-training loss against dense baselines at matched active parameter count, demonstrating practical utility.

weaknesses
The paper's empirical validation of loss curve collapse (Finding 1.2) is restricted to width, expert count, and expert size, and excludes depth. Figure 17 acknowledges degraded load balancing when scaling over depth and authors attribute to constant-HP tuning without further investigation. This interference is a regression to the state of the art and should be discussed more thoroughly.

The set of assumptions seem quite restrictive in practice: in real life the number of active experts is fixed, and this shows to break transfer (fig 10). The auxiliary-loss-free bias update for load balancing is adopted without ablation. Furthermore, the parameterization only addresses learning rate and initialization scale, leaving other practically important hyperparameters: batch size, Adam betas, and LR scheduling.

It is unclear to what extent the DMFT analysis applies to actual training settings. The scaling rules are derived for Adam via approximation with SignGD. The DMFT justification assumes gradient flow, with the authors noting it "can be easily relaxed to discrete time SGD". The relaxation is not performed and does not seem trivial.

The presentation has several issues: typos, missing words, and the conclusion is delegated to the appendix rather than appearing in the main text, leaving the main text feeling truncated and lacking synthesis and a final discussion.

The novelty of the work is diminished by the concurrent work Malasnicki et al. , which derives LR transfer rules for MoE models under width scaling. The present paper's scope is broader (width, depth, expert count, expert size) and with initialization transfer. The authors acknowledge the concurrent work, but don't clarify whether the two approaches agree on scaling rules for shared dimensions. A direct comparison would help measure the novelty and the impact of the contribution more clearly.

---

> ### Author Rebuttal · Authors · 2026-03-26
>
> We deeply appreciate the insightful comments. Regarding writing, we will work our best to clean up typos and presentations. In the next version (with added page limit), we will incorporate the feedback into a detailed conclusionary section on limitations, theory vs empirics, and future works. Below we address technical comments:
>
> **1. Necessity and actual impact of DMFT analysis?**
>
> The DMFT analysis is not necessary to work out an initial guess based on dimensional analysis for init, multipliers, or LRs. However, dimensional guesses are neither always correct nor guarantees HP transfer. The DMFT analysis (1) establishes consistency and correctness of initial guesses via the existence of a single joint scaling limit (experts, expert size, depth, etc), and (2) provides further non-trivial insights, e.g. the somewhat surprising HP transfer (model dynamics invariance) over different FFN ratios.
>
> **2. Gradient flow to Adam?**
>
> Gradient flow analysis is more theoretically tractable and is used to make predictions in a variety of prior literature on HP transfer that extends reliably on both SGD and Adam. While prior works have extended DMFT directly to (S)GD (and we believe this can be done in our setup as well), rigorously studying Adam is challenging due to per-entry norm. A full DMFT analysis of Adam or other practical optimizers will be very interesting!
>
> **3. Depth scaling is unsatisfying. To what extent does MoE interfere w/ CompleteP?**
>
> Thank you for raising this point and we will clarify further in the next version. Due to character limits, we point to our response to Reviewer YUNY. In short, we adopt dense/CompleteP depth recipes, which also enable reliable HP transfer for MoE models, and we found nothing out of line empirically versus dense/CompleteP. While a close examination reveals internal load imbalance, we explicitly identify them in early layers (a known issue in literature) and leave detailed investigations to this future work.
>
> **4. Architecture restriction, aux loss**
>
> You are certainly correct that our empirics and theory are architecture-specific. In general, there is no widely agreed upon setup for how to implement MoEs and many variants exist. The main goal of ours is therefore to study  **one** architecture (employed in frontier open source LLMs by DeepSeek, Kimi, and Arcee) with theoretically backed scaling recipes that empirically also (1) reliably scale up with HP transfer; (2) achieve good performance with minimal tuning.
>
> While we believe that our approach can likely be adapted to alternative architectures (see also our response to Reviewer 2AVh), an aux loss to solely enforce load-balancing appears more difficult and doesn't fit into HP transfer frameworks perfectly. This is because an aux. loss (or most global regularizers) carries with it many complex questions of how one ought to balance the tradeoff of gradients for each parameter group between the primary data-fitting objective and the regularizer (e.g. should layer-1 self-attention weights be equally responsible for load-balancing in layer-10), and how to evaluate models with different HPs. It is not clear how to do this in a principled and rigorous manner, and it appears that fine-grained practitioner-specific desiderata will lead to drastically different scaling analysis and training recipes. We discuss this briefly in Apdx D.2 and will further clarify in our revision.
>
> **5. Fixed n_act?**
>
> We agree that at inference time, one might prefer to keep a fixed number of experts activated. However, our theoretical analysis and empirics suggest constant sparsity is necessary for HP transfer. We briefly discussed this issue in Sec 3.2 (beyond which, another problem is that picking the top-1 out of n experts makes the average logit scale like log(n) which leads to additional corrections that likely change during training). While a constant sparsity may sometimes be sub-optimal for inference hardware, our practical recommendation is to (a) start with a fixed target sparsity and scale up granularity; or (b) fix an expert configuration and scale up width/depth; or (c) apply distillation after training a model with more active parameters.
>
> **6. Novelty to Malasnicki et al.**
>
> As discussed in our paper, we acknowledge Malasnicki et al. as a valuable reference that studies width scaling which matches our results for widths. However, we believe that our analysis goes beyond in completeness and rigor both theoretically (beyond scale heuristics, we justify DMFT limiting dynamics and further insights) and empirically (beyond checking LR transfers across width for one setting, we cover more/all model scaling dimensions, discuss detailed design choices and requirements for well-defined transfer desiderata, and report internal model behaviors with empirical findings for proper scaling).
>
> **We hope the above addresses your concerns. We welcome remaining questions and kindly ask that you consider raising your score if they are resolved.**

---

> > ### Author Rebuttal · Reviewer_BLe8 · 2026-04-07
> >
> > We thank the authors for the detailed answers. We hope to see the adequate improvement in presentation. The authors clarified with precision the scope and the impact of their work, and we decide not to update our scores because of the unpractical assumptions necessary to make the experiments positive.

---

> > > ### Author Response · Authors · 2026-04-07
> > >
> > > Thank you for suggestions towards presentation and for acknowledging our response regarding scope and impact. As the discussion period concludes shortly, we want to briefly further clarify our assumptions mentioned in the review:
> > >
> > > **1. DMFT assumptions**
> > >
> > > Our assumptions in the DMFT analysis are in line with a number of prior works used for HP transfer (e.g. [1] and references therein). Using our techniques, one can write down DMFT equations for SGD/SignGD, which will be qualitatively similar to the existing analysis.
> > >
> > > While DMFT analysis for Adam is open, the SGD/flow Theory -> Adam Empirics pipeline in predicting HP transfer has been empirically tested to be reliable and useful in many prior works at large scale (e.g. [2]).
> > >
> > > **2. Fixing sparsity**
> > >
> > > Our theory supports but doesn't require a scaling number of experts: we show that training dynamics solely depend on sparsity (and does *not* depend on granularity). Despite theoretical non-transfer across sparsity, we gave (see prior response) multiple practical ways one can effectively achieve the same objective (to train large model with fixed active experts and many total experts).
> > >
> > > From the scaling perspective, it is also not obvious that active experts bottleneck pretraining at a hardware level. In practice, communication cost in pretraining hardware often requires lower-bounded sparsity (see Sec 5.3 and Table 4 in [3] as well as Sec 3.2 in [4]), which is exactly our studied setting (fixing sparsity from base model).
> > >
> > > **3. Aux-loss**
> > >
> > > Our main point is that analyzing a global auxiliary loss requires much more practitioner-specific assumptions (e.g., desiderata in balancing gradients in different layers, desiderata in comparing different models), and that blanket application is not theoretically justified (in fact, aux-loss is **not** globally minimized with balanced routing). Furthermore, even the empirical application of aux-loss requires scrutiny (e.g. batch-wise vs sequence-wise balancing achieve different objectives, see Fig.9 in [4]). While still not a perfect solution, our choice of aux-loss-free balancing is both empirically popular as a replacement and rids us of having to make much more fine-grained assumptions (e.g. only specific biases receive balancing gradients, and sequence vs batch imbalance losses coincide).
> > >
> > > **While we acknowledge the existence of our limitations, we politely maintain that our experiments are conducted at highly practical setups via training realistic models, and that our theoretical assumptions are in line with vast lines of prior literature.**
> > >
> > > [1] Depthwise Hyperparameter Transfer in Residual Networks
> > >
> > > [2] Don't be lazy: CompleteP enables compute-efficient deep transformers
> > >
> > > [3] Step-3 is Large yet Affordable: Model-system Co-design for Cost-effective Decoding
> > >
> > > [4] DeepSeek-V3 Technical Report

---

### Official Review · Reviewer_YUNY · 2026-03-16

**Soundness:** 3
**Presentation:** 3
**Significance:** 3
**Originality:** 3
**Overall Recommendation:** 4
**Confidence:** 3

**Summary:**

This paper studies hyperparameter transfer in transformer MoEs when scaling width, depth, number of experts, and expert hidden size. The main proposal is a new parameterization for MoE models, motivated by a DMFT analysis, with the goal of making learning rates and initialization settings transfer reliably across scales. Empirically, the paper shows transfer from relatively small models to much larger ones, and also explores the tradeoff between using more experts versus larger experts.

**Compliance With Llm Reviewing Policy:**

Affirmed.

**Key Questions For Authors:**

-Depth scaling looks weaker than the other scaling directions. Do the authors see this as a limitation of the parameterization, or more of an issue with the current experimental setup?
-Can the authors clarify more explicitly which parts of the scaling rule are directly supported by the DMFT analysis, and which parts are more heuristic?

**Limitations:**

The limitations are only partly discussed. I think the paper should more clearly acknowledge the dependence on tuned constants, the weaker depth-scaling evidence, and the fact that the experiments are still limited to the specific architectures and training regimes considered here.

**Strengths And Weaknesses:**

I think this is a solid paper on an important and timely problem. Hyperparameter tuning for MoEs is expensive and messy, so a method that makes transfer more reliable would be genuinely useful.

The main strengths are that the paper tackles a practical problem, the proposed scaling rules are clearly stated, and the empirical section is fairly extensive. The experiments cover several scaling axes rather than just width, which is a real plus. I also liked that the paper does not stop at small-scale sweeps, but also tests whether hyperparameters found on smaller models can be reused for larger and longer runs. The theory side is also a positive: the DMFT analysis gives the work more depth than a purely empirical recipe paper.

My main concern is that some of the practical story seems a bit less clean than the headline claim suggests. In particular, the method still seems to rely on tuning constant multipliers for some parameter groups, so the transfer is not completely automatic in practice. Also, the depth-scaling results seem less convincing than the width/expert-scaling results, and I think that limitation should be stated more clearly in the main paper. On the theory side, the analysis is interesting, but I did not check all derivations carefully, so I cannot fully judge how strong the theoretical justification is.

In terms of presentation, the paper is generally clear and well organized, but a bit dense. I think the authors could do a better job separating what is theoretically justified, what is heuristic, and what is mainly an empirical observation. A clearer summary of the exact scope of transfer would also help.

Overall, I think the paper is meaningful and reasonably original. It is not a huge conceptual leap, but extending hyperparameter transfer ideas carefully to sparse MoEs, with both theory and experiments, is a worthwhile contribution.

---

> ### Author Rebuttal · Authors · 2026-03-26
>
> We deeply appreciate the insightful comments. In the next version (with the added page limit), we will incorporate the feedback into a detailed conclusionary section on assumptions and limitations, theory vs empirics, and future works. Below we address technical comments. We hope they address your concerns and welcome any remaining questions:
>
> **1. Parameter group constant-level tuning.**
>
> Thank you for bringing this up. Indeed, experimentally we needed to tune constant multipliers in the small (50M) **base model only** to ensure training stability. We want to note that:
>
> 1. Making constant-scale tuning in different types of weight matrices is a standard and studied practice for stability and performance. Even in dense models, when training is inherently more stable, performance-driven implementations still make efforts to tuning and report benefits. Our message is that for MoEs, a minimal level of tuning is not only better for performance but also necessary for stability.
>
> 2. Empirically, it is also more reasonable to scale up from a base model (on which tuning is cheap and efficient) that is both stable and performant at a small scale, as base model misalignments/suboptimality could likely get exaggerated when scaling up (even if the scaling recipe is correct).
>
> 3. In our experiments we only did a simple coordinate search over the base model for a reduced number of parameter groups, which suffices to our purpose. Our point is that simple tuning for base model stability is already sufficient for reliable scaling with our recipe, and searching a SOTA base model is not required.
>
> **2. Depth scaling seems less convincing, in regard to loss collapse and load imbalance.**
>
> We agree that depth scaling behaves differently from width or other dimensions, both theoretically and empirically, and some aspects of our scaling may appear less stable. Thank you for bringing this up and we will make further explicit clarifications in the next version.
>
> 1. Our depth-parameterization is adopted from prior work on dense models (CompleteP) and we (crucially) obtain optimal LR and init. transfer across depth in MoEs. Furthermore, DMFT justifies the scaling limit is indeed uniquely defined (invariance across depth).
>
> 2. We note that while theory predicts $L\to \infty$ behavior, realistic depths (e.g. base $L=8$) are much smaller and finite-depth effects are possible. In fact, we notice that even in dense models (CompleteP), small depths induce misalignments in early loss profiles.
>
> 3. Load-imbalances at large depth happen ONLY during the first few layers, while deeper layers still always balance perfectly. Imbalanced routing in early layers is a known issue in open source LLMs such as DeepSeekMoE, DeepSeek V3, and Arcee-Trinity, which leads to their using dense blocks in early layers before MoEs in deeper layers. While recent works (e.g. Quantile Balancing) have proposed ways to mitigate this issue, they fall outside the scope of our current work.
>
> **3. What is DMFT-justified, what is only heuristics, and what is mainly empirical**
>
> Our theoretical approach to scaling exponents is motivated by first using dimensional scale analysis to heuristically estimate how to scale everything. Then, we use a (slightly simplified) MoE model in DMFT to rigorously justify a consistent and well-defined limit as well as derive additional theoretical insights (e.g. an FFN-ratio independent scaling limit exists). In this sense, all proposed scaling recipes (exponent laws for LR and init) derived by heuristics are justified by DMFT, albeit in a simplified model (asymptotic mean-field limit) and training setting (population descent, constant training horizon, simplified optimizer). Our simplifications in DMFT modelling is in line with a variety of prior works using DMFT to justify HP transfer in deep resnets, self-attention layers, and others.
>
> Empirical assumptions that aren't covered by theory are therefore finite-dimensional behavior (e.g. verifying that base $n=512$ suffices for HP transfer towards the infinite width limit) and optimization practicalities (e.g. constant multipliers, potentially variable training horizon, finite batch size, LR-scheduling, and Adam dynamics), all of which tested to show our recipe's practicality. We thank the reviewer for pointing this out, and we will make sure to clarify this in our next version.
>
> **We hope the above addresses your concerns. We welcome any remaining questions and kindly ask that you consider raising your score if they are resolved.**

---

> > ### Author Rebuttal · Reviewer_YUNY · 2026-04-01
> >
> > The answers to a few questions are still a bit interpretative.
> >
> > However, I recommend acceptance.

---

> > > ### Author Response · Authors · 2026-04-03
> > >
> > > Thank you for your valuable review and feedback comments, which we will adopt to improve the paper in the revision.
> > >
> > > To follow up quickly on the DMFT vs. heuristics question, we will try to provide a more concrete explanation of the utility of the DMFT and its connection to the experiments.
> > >
> > > 1. The rules for how to set scaling exponents in learning rates and initializations can be identified heuristically  by verifying that the entry-wise updates in hidden states of the networks are $\Theta(1)$ with respect to scaling variables. This type of analysis is common in applied muP papers.
> > >
> > >
> > > 2.  However, to *prove* that these rules actually lead to models converging to a stable infinite limit, we need to verify that there is a well defined dynamical system that no longer depends on exact scaling variables $n_{embd}, n_{exp}, n_{act}, L$. For instance, we use the DMFT to show that there is indeed a limit where $n_{act}/n_{exp} = \kappa$ that does not depend on $n_{embd}, n_{exp}, n_{act}, L$ but only on $\kappa$. This implies that dynamics should be matching across all $n_{exp}, n_{act}$ if $\kappa$ is fixed. We are able to provide the exact analysis of the limit under SGD / gradient flow, and expect the result to be similar for other optimizers such as muon or Adam.
> > >
> > >
> > > 3. The theory also indicates that the limiting dynamics, though they depend on $\kappa$, do *not* depend on the FFN ratio (expert size). This provided theoretical support for the empirical HP transfer plots across FFN ratio, which *cannot* be verified through just heuristic dimensional analysis.

---

### Decision · Program_Chairs · 2026-04-30

**Decision:**

Accept (regular)

**Comment:**

This paper studies hyperparameter transfer in MoE Transformers and proposes a scaling parameterization, motivated by dynamical mean-field theory, to enable consistent transfer of learning rate and initialization across multiple scaling dimensions.

The reviews converge on a strong endorsement of the paper’s technical soundness. There is clear agreement that the work tackles an important and insufficiently explored problem, and the empirical evidence showing reliable transfer from small to larger models is viewed as a particularly compelling contribution.

Main Weaknesses: The method still requires base-model constant tuning, limiting full automation. Depth scaling is less convincing and insufficiently analyzed. The approach relies on restrictive assumptions and specific architectures, raising concerns about generality. The theory uses simplifying assumptions that may not fully match practical training.

Overall, the paper is technically sound and makes a meaningful contribution to hyperparameter scaling for MoE models. While the novelty is incremental and the applicability is somewhat constrained, the combination of empirical validation and theoretical motivation provides sufficient value.